# Stress-induced brain responses are associated with BMI in women

Anne Kühnel [1,2,3✉], Jonas Hagenberg [2,3,4], Janine Knauer-Arloth [2,4], Maik Ködel[2], Michael Czisch[5], Philipp G. Sämann[5], BeCOME working group*, Elisabeth B. Binder [2,6✉] & Nils B. Kroemer [1,6,7]

Overweight and obesity are associated with altered stress reactivity and increased inflammation. However, it is not known whether stress-induced changes in brain function scale with BMI and if such associations are driven by peripheral cytokines. Here, we investigate multimodal stress responses in a large transdiagnostic sample using predictive modeling based on spatio-temporal profiles of stress-induced changes in activation and functional connectivity. BMI is associated with increased brain responses as well as greater negative affect after stress and individual response profiles are associated with BMI in females ($p_{perm} <$ 0.001), but not males. Although stress-induced changes reflecting BMI are associated with baseline cortisol, there is no robust association with peripheral cytokines. To conclude, alterations in body weight and energy metabolism might scale acute brain responses to stress more strongly in females compared to males, echoing observational studies. Our findings highlight sex-dependent associations of stress with differences in endocrine markers, largely independent of peripheral inflammation.

[1] Section of Medical Psychology, Department of Psychiatry and Psychotherapy, Faculty of Medicine, University of Bonn, Bonn, Germany. [2] Department of Translational Research in Psychiatry, Max Planck Institute of Psychiatry, Munich, Germany. [3] International Max Planck Research School for Translational Psychiatry (IMPRS-TP), Munich, Germany. [4] Institute of Computational Biology, Helmholtz Zentrum Munich, Neuherberg, Germany. [5] Max Planck Institute of Psychiatry, Munich, Germany. [6] German Center for Mental Health, Tübingen, Germany. [7] Department of Psychiatry and Psychotherapy, Tübingen Center for Mental Health (TüCMH), University of Tübingen, Tübingen, Germany. *A list of authors and their affiliations appears at the end of the paper. ✉email: akuehnel@uni-bonn.de; binder@psych.mpg.de

Stress is an everyday occurrence, but prolonged exposure to stress increases the risk for a number of negative health outcomes, including for metabolic and cardiovascular disease[1]. Chronic stress has been associated with a heightened risk for obesity[2] which is also associated with cardiovascular events[3–5] as well as dysregulations of energy metabolism[6] and the immune system[7]. Obesity is also related to altered functioning of the hypothalamus-pituitary axis (HPA)[8] as indicated by, for example, reduced baseline HPA activity levels[9,10] and possibly stronger acute endocrine stress reactivity[11]. Consequently, this possible bidirectional link between stress and obesity may be of high relevance for the pathophysiology of stress-related disorders.

Although chronic stress is a well-known risk factor for obesity, it has not been conclusively resolved whether there are replicable differences in the acute stress response with obesity. To date, most studies have focused on altered endocrine stress responses in obesity. While there is evidence that acute cortisol responses to stress are higher in obesity[11], such associations are inconsistent across studies[12–14]. Likewise, higher body mass index (BMI) is consistently associated with a blunted stress-induced cardiovascular response[15,16]. BMI-dependent differences in subjective responses to acute stress are less well characterized, although one potential link between stress and increased food intake is compensatory eating in response to negative emotions[17]. Notably, sex-dependent associations of stress reactivity and obesity are an important mechanism to better understand sex differences in the prevalence of obesity[18,19] and its relation to mental[20] as well as metabolic disorders[21,22]. First, there are sex differences in stress responses as females have shown increased subjective but blunted endocrine responses[23,24]. Likewise, neural stress responses differ between males and females[25–27], including associations between stress-induced brain responses and subjective stress experiences[28,29]. Second, sex hormones regulate endocrine stress responses[30] and energy metabolism[20], substantiating potential sex differences in the interplay between stress and BMI. Taken together, there is preliminary evidence for changes in acute stress reactivity in overweight and obesity, but little is known about neural changes or potential sex differences in humans.

In addition to changes in endocrine and cardiovascular systems, overweight and obesity are also characterized by increased inflammation[31–33]. Baseline levels of cytokines, such as macrophage migration inhibitory factor (MIF) and interleukin 6 (IL-6), were associated with stress-induced cortisol responses[34,35], highlighting the interdependence of the immune and endocrine system in orchestrating stress responses[36]. This interdependence is further highlighted as increased inflammation induced by vaccination[37], in depression[38], and obesity[39,40] has been linked to changes in functional connectivity (FC) in brain networks that are also implicated in stress reactivity[41]. Mirroring sex differences in stress reactivity and obesity, the immune system markedly differs between males and females[42]. Obesity is strongly associated with increased inflammation in females[43,44] and sex hormones have been proposed to explain the sex-dependent interplay of obesity, stress reactivity, and the immune system[30,45]. To conclude, obesity is linked to inflammation in a potentially sex-dependent manner and interactions with the endocrine and the immune system may tune acute stress responses, potentially mediating the effects of obesity on stress.

Despite the recent progress in delineating the link between the immune system and responses to stress, it is not yet understood by which mechanism obesity may contribute to altered stress reactivity. Here, we investigated the modulating effects of BMI on subjective, autonomous, endocrine, and neural stress reactivity. To this end, we first investigated stress-induced changes in brain responses (i.e., activation changes). Moreover, obesity has been robustly related to alterations in FC[46]. We have previously related stress-induced dynamic FC trajectories within a putative stress network with negative affectivity, a risk factor for mood and anxiety disorders that are often comorbid with obesity[41]. Therefore, we derived dynamic FC and activation trajectories and used cross-validated elastic nets to evaluate robust associations of these imaging features with interindividual differences in BMI. We then evaluated whether BMI-associated increases in baseline cortisol or inflammation markers contributed to altered neural stress reactivity in obesity. By unraveling sex-specific associations of BMI with acute stress reactivity, we shed new light on the interrelation of stress and obesity in females.

## Results

**Higher stress-induced negative affect in high BMI participants.** As reported in Kühnel et al.[41], the task elicited a robust multi-level stress response (Fig. 1). This stress response was indicated by increases in heart rate (during stress: unstandardized estimate ($b$) = 6.7, $t(159)$ = 13.0, $p < 0.001$), negative affect (after stress (T6): $b$ = 7.7, $t(183)$ = −12.6, $p < 0.001$) and decreases in positive affect (after stress (T6): $b$ = −2.3, $t(183)$ = −8.0, $p < 0.001$). Cortisol increased in response to stress (T6 after stress, ~16 min after stress onset) in participants not showing an increase in cortisol to the blood drawing procedure ($b$ = 0.4, $t(178)$ = 2.6, $p = 0.011$). After stress, heart rate recovered but not to baseline levels (after stress during recovery: $b$ = 0.88, $t(159)$ = 2.1, $p = 0.034$). Moreover, after a 30-min break, negative affect (T8: $b$ = −1.3, $t(183)$ = −4.7, $p < 0.001$), but not positive affect (T8: $b$ = −1.1, $t(183)$ = −2.5, $p = 0.014$), had recovered. On the neural level, the task led to robust stress-induced increases in activation in the visual and parietal cortex as well as decreases in posterior cingulate cortex (PCC), angular gyrus, insula (posterior and anterior), supplementary motor area (SMA) and dorsomedial prefrontal cortex[41,47,48].

**BMI is related to stress reactivity and cytokine concentrations.** Next, we evaluated the effect of sex and BMI on stress reactivity and cytokine levels. On the subjective level, a multivariate regression (MV) including changes in negative and positive affect at both timepoints after stress induction revealed that a higher BMI was associated with greater stress-induced changes in affect ($p_{MV} = 0.019$). Specifically, a higher BMI was related to more negative affect after the task ($b$ = 1.48, $t(182)$ = 2.00, $p = 0.047$, $N_{females}$ = 120, Fig. 2) and after the 30-min rest period ($b$ = 1.18, $t(182)$ = 2.35, $p = 0.019$), relative to the baseline. This association was significant in females at the later time point (T6: $b$ = 1.7, $t(115)$ = 1.84, $p = 0.069$; T8: $b$ = 1.3, $t(115)$ = 2.28, $p = 0.025$) but not in males (T6: $b$ = −1.1, $t(66)$ = −1.20, $p = 0.24$; T8: $b$ = 0.6, $t(66)$ = 0.76, $p = 0.44$; Fig. 2b), but the interaction between sex and BMI did not reach significance (T6: $t(182)$ = −1.68, $p = 0.095$; T8: $t(182)$ = −0.44, $p = 0.66$). In contrast, higher BMI was not associated with stress-induced changes in heart rate, or cortisol concentrations ($p$s > 0.10, Fig. 2a, Table S4) and subjective, cardiovascular, and endocrine stress responses did not differ between males and females ($p$s > 0.15; Fig. 2a, Table S2).

In line with a link between altered stress reactivity and increased inflammation in overweight and obesity, a higher BMI was associated with increased peripheral cytokine levels ($p_{MV}$ = 0.006), including increased high sensitivity C-reactive protein (hsCRP), interleukin (IL)-1 receptor antagonist (IL-1RA), tumor necrosis factor (TNF-alpha), IL-16, and soluble IL-6 receptor (sIL-6R) among others (full list of partial correlations with $p < 0.05$: Fig. S2). Cytokine levels were more strongly associated with BMI in females compared to males (BMI × Sex $p_{MV}$ = 0.018). In contrast, baseline cortisol (measured in plasma samples at the beginning of an independent session ~8 am) was lower in

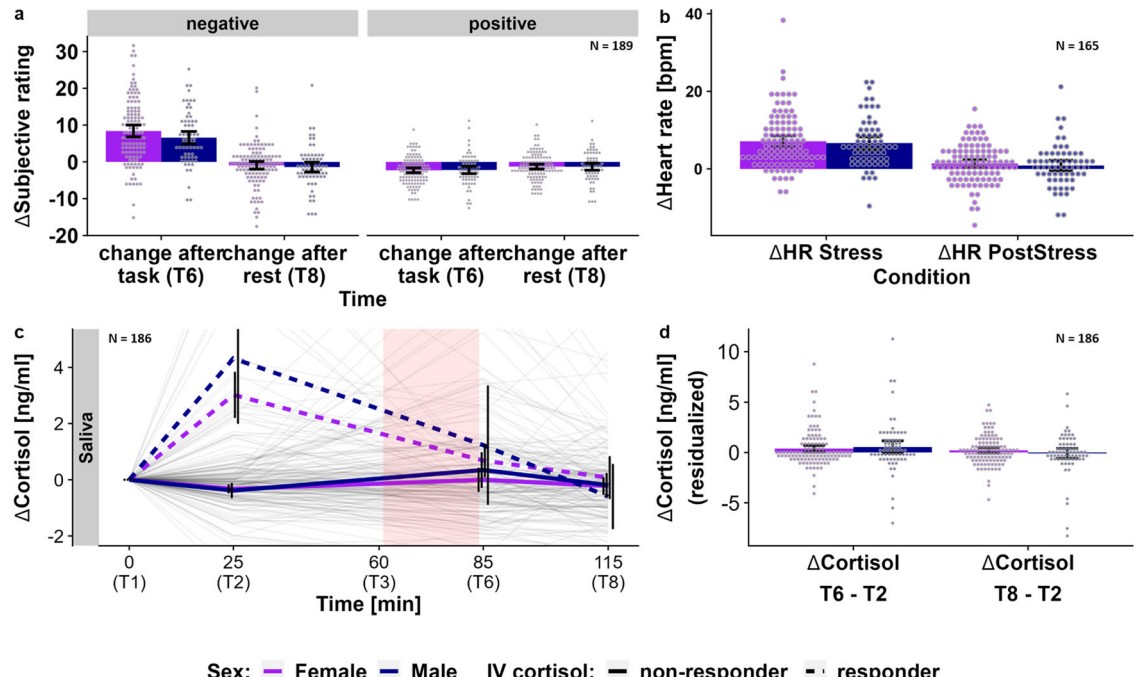

**Fig. 1 The stress task induced a similar stress response on the endocrine, subjective, and cardiovascular level in females and males. a** Stress-induced increases ($n = 189$), relative to T2, (T6: $b = 7.7$, $t(183) = -12.6$, $p < 0.001$) in negative affect recover below baseline levels after stress (T8: $b = -1.3$, $t(183) = -4.7$, $p < 0.001$). At the same time, positive affect decreases (T6: $b = -2.3$, $t(183) = -8.0$, $p < 0.001$) but does not fully recover (T8: $b = -1.1$, $t(183) = -2.5$, $p = 0.014$). **b** Stress induces increases in heart rate ($n = 165$, $b = 6.7$, $t(159) = 13$, $p < 0.001$). During PostStress, heart rate decreases again ($b = -5.8$, $t(159) = -12.4$, $p < 0.001$) but does not fully recover ($b = 0.9$, $t(159) = 2.1$, $p = 0.033$). **c** Stress induces a cortisol response ($n = 186$) in participants not already reacting to the placement of an intravenous catheter ("non-responder") compared to the pre-task cortisol measurement (T2). Cortisol levels recover close to baseline levels after the 30-min break. Thin lines depict individual cortisol trajectories; thick lines show group averages. The shaded area shows the timing of the stress task. **d** Changes in cortisol ($N = 186$) directly after the task (T6 – T2) and after the 30-min rest (T8 – T2) do not differ between males and females. Values in **a**, **b**, and **d** show residualized (age, sex, IV-cortisol response, diagnosis status) averages and confidence intervals (95% CI) for males and females separately. Source data are provided in the Supplementary Data 1.

participants with high BMI (rho(142) = −0.17, $p = 0.015$) and did not differ between sexes (BMI × Sex $t(142) = 0.46$, $p = 0.64$, rho$_{female}$(83) = −0.27, $p = 0.003$; rho$_{male}$(53) = −0.15, $p = 0.22$).

Similar to negative affect, higher BMI was associated with stronger stress-induced decreases of BOLD responses in the posterior insula (L: $p_{FWE} < 0.001$, $k = 293$, R: $p_{FWE} = 0.041$, $k = 143$) and a midbrain cluster including the substantia nigra ($p_{FWE} < 0.001$, $k = 390$) as well as increased BOLD responses in the precuneus/superior parietal lobe ($p_{FWE} = 0.025$, $k = 145$, Fig. 3, for unthresholded maps, see https://neurovault.org/collections/NABGNECT/). Notably, within the pre-defined stress-related network, females showed a more negative correlation with BMI in the hippocampus (Sex × BMI: $t_{max} = 4.26$, $p_{SVC.Hippocampus} = 0.008$; $p_{SVC.StressNetwork} = 0.088$). Moreover, BMI-associated changes in stress responses across ROIs (Shen atlas[49]) calculated separately for males and females were not spatially correlated ($r(266) = 0.05$, $p = 0.38$, Fig. S5). To evaluate potential sex effects, we conducted post hoc regression analyses (including BMI × Sex interactions as well as separately for males and females) on average beta values extracted from ROIs[49] overlapping the posterior insula and substantia nigra. While the interaction of sex and BMI did not reach significance (posterior insula R: $t(183) = -1.77$, $p = 0.071$, SN: $t(183) = -0.94$, $p = 0.36$), associations were numerically higher in females (Fig. 3b, for details see SI).

**Stress-induced trajectories of brain activation are related to BMI in females**. To associate stress-induced changes in brain responses with BMI and derive individual predictions for unseen data, we used cross-validated predictive modelling with elastic

nets. This prediction captures the variance explained by spatio-temporal profiles of stress-induced changes in brain activation and FC, which we have previously shown to recover negative affectivity beyond conventional analyses[41]. The model successfully captured BMI based on activation ($\Delta R^2 = 0.07$, $p_{perm} = 0.0032$, Fig. 4a) and including FC did not improve prediction (for predictive accuracies see Fig. 4b, observed (yellow) vs. permuted error bars). Of note, the BMI predicted by the elastic net model covered a smaller range than the observed BMI, as predicting values close to the mean is less penalized, leading to shrinkage[50] while the relative information between participants is largely unaffected. BMI was predicted by higher activation of the anterior hippocampus, ventromedial prefrontal cortex, and dorsal anterior cingulate cortex (dACC) as well as lower activation of the posterior insula and posterior hippocampus mirroring whole-brain associations (Fig. 4c and overlaid on the corresponding ROI 4e, selected features are weights ≠ 0). Notably, features from the posterior insula, hippocampus and dACC contributed most to the prediction as evaluated by excluding the corresponding feature in the prediction (Fig. 4d). Crucially, the elastic net only performed better than chance in females (females: $r(118) = 0.26$, $p = 0.005$; males: $r(68) = -0.05$, $p = 0.66$, Sex × BMI: $t(186) = 2.7$ $p = 0.03$, Fig. 4a) and re-training the model only in females further improved the accuracy ($\Delta R^2 = 0.11$; $p_{perm} = 0.002$, compared to a model including only confounding variables, Fig. 4f), although features were similar (Fig. S4). Moreover, using models trained on females to predict BMI in males was not successful (and vice versa), indicating that neural stress responses reflective of BMI differ between sexes.

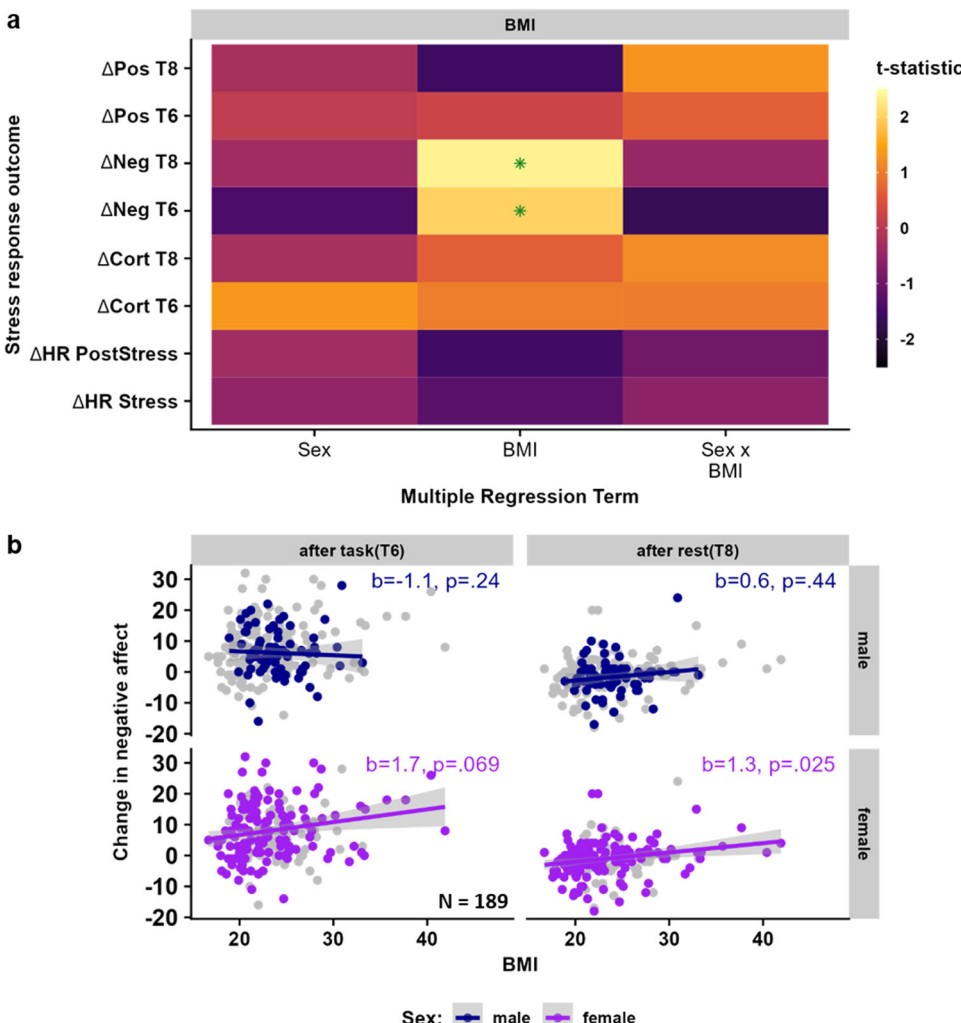

**Fig. 2 Stress-induced increases in negative affect are larger in participants with a high body mass index (BMI). a** Estimates for the effects of sex, BMI, and their interaction from regression models of multi-level stress responses (subjective response: $n = 189$ participants, cardio-vascular response: $n = 165$ participants, and cortisol response: $n = 186$ participants). Estimates are $t$-values from linear multiple regressions adjusted for linear effects of age, presence of a psychiatric diagnosis, and cortisol response to intravenous catheter placement. Green asterisks indicate significant results ($p < 0.05$). **b** Scatterplots showing associations of $n = 186$ participants between BMI and stress-induced negative affect after the task (both sexes: $b = 1.48$, $t(182) = 2.00$, $p = 0.047$) and after the following 30-min rest (both sexes: $b = 1.18$, $t(182) = 2.35$, $p = 0.019$) separated for males and females to depict potential sex differences. Associations with BMI were significant in $n = 120$ females (T8), but not in $n = 69$ males. While models account for covariates, the data is shown unadjusted in the scatterplots. $\Delta$Cort T6 = Cortisol increase after the end of the task (T6) compared to baseline (T0). Shaded areas depict 95% confidence intervals of the association of unadjusted data. $\Delta$Cort T8 = Cortisol increase after rest (T8) compared to baseline (T0). $\Delta$HR PostStress = Difference in heart rate between task-blocks in the PostStress and PreStress condition, $\Delta$HR Stress = Difference in heart rate between task-block in the PostStress and PreStress condition, $\Delta$Neg T6 = Difference in state negative affect directly after the task (T6) compared to before the task (T3), $\Delta$Pos T6 = Difference in state positive affect directly after the task (T6) compared to before the task (T3). Source data are provided in the Supplementary Data 1.

**Baseline cortisol, but not cytokines, are associated with elastic-net predicted BMI.** We hypothesized that sex-dependent associations between BMI and cytokines might explain the sex-dependent associations between BMI and stress responses. To this end, we evaluated if BMI-associated cytokines were correlated with the BMI predicted by the multivariate elastic net model versus the residual BMI not predicted by stress-induced brain responses. We reasoned that associations of cytokines with predicted BMI would indicate shared variance, whereas associations with residual BMI would indicate that variance attributable to cytokines is not captured by the predictive model based on stress-induced changes in brain function. Cytokines related with BMI (Fig. 5a: multiple regression estimates for the effect of sex, cytokine concentration and their interaction on BMI) were only associated with the residual BMI and not the predicted BMI

(Fig. 5a), suggesting that such differences in inflammation do not account for BMI-associated differences in stress-induced brain responses. In contrast to cytokines, reduced baseline cortisol levels were associated with a higher model-predicted BMI in a multiple regression model (BMI: $b = -0.31$, $t(142) = -2.30$, $p = 0.023$; BMI × Sex: $t(142) = 1.73$, $p = 0.086$, regression estimates Fig. 5a, scatterplot Fig. 5b).

## Discussion

Stress has been related to an increased risk for overweight and obesity, and differences in BMI are associated with altered acute stress reactivity. However, the contribution of increased peripheral inflammation and changes in the HPA axis with higher BMI are not yet well understood. Here, we demonstrate that a higher

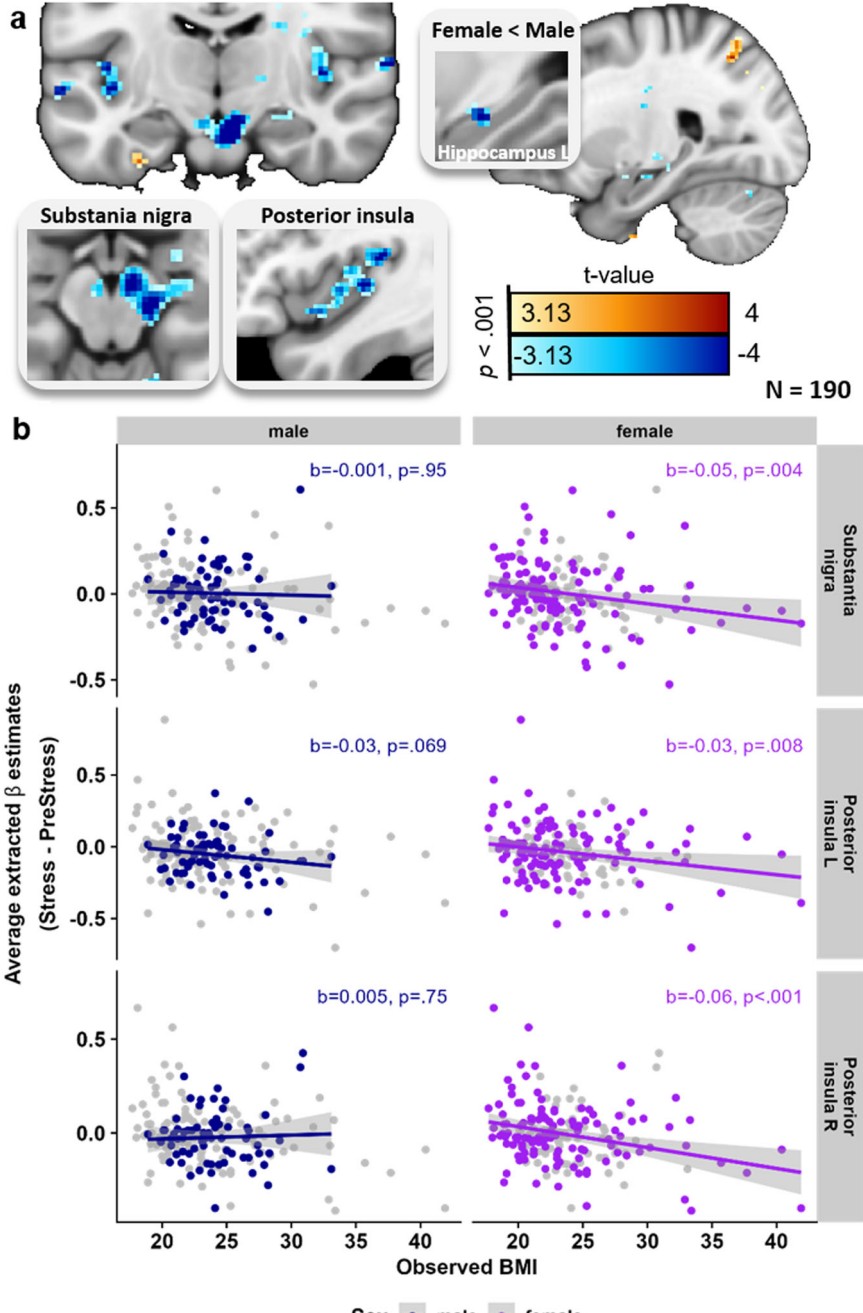

**Fig. 3 Stress-induced deactivations in the posterior insula and substantia nigra are stronger with increasing BMI. a** Whole-brain regression analyses ($n = 190$) show associations between body mass index (BMI) and stress-induced (Stress – PreStress) activation changes. Higher BMI is associated with increased (warm colors) stress-induced activation in the superior parietal lobe/precuneus and decreased (cool colors) activation in the substantia nigra and posterior insula. Voxel-threshold for display: $p < 0.001$, $t > 3.13$. **b** Extracted beta estimates (average across the region of interest, ROI) from corresponding ROIs defined in the Shen atlas[49] are negatively associated with BMI. Regression weights and significance values are derived from separate multiple regressions for $n = 120$ females and $n = 70$ males (accounting for confounds and data is shown unadjusted for the visualization). Shaded areas show 95% confidence intervals for the associations of unadjusted data. Associations with BMI are only significant in females. BMI = body mass index, L = left, R = right. Source data are provided in the Supplementary Data 1.

BMI was reflected in distinct activation trajectories across stress phases derived from the posterior insula, the dACC, and the hippocampus (anterior and posterior) within a pre-defined network. Accordingly, participants with a higher BMI showed stronger stress-induced brain responses in the posterior insula that were primarily driven by observed alterations in females. Additionally, whole-brain analyses revealed stronger stress-induced responses in the substantia nigra and the parietal cortex of participants with higher BMI, suggesting that a more extensive stress-associated network is affected. Notably, associations of BMI with stress-induced changes were not spatially correlated in males and females, pointing to sex-specific associations (see the corresponding maps on neurovault (https://neurovault.org/collections/NABGNECT/) that can be used in future studies). Crucially, different levels of peripheral cytokines did not account for altered stress reactivity in the brain as they

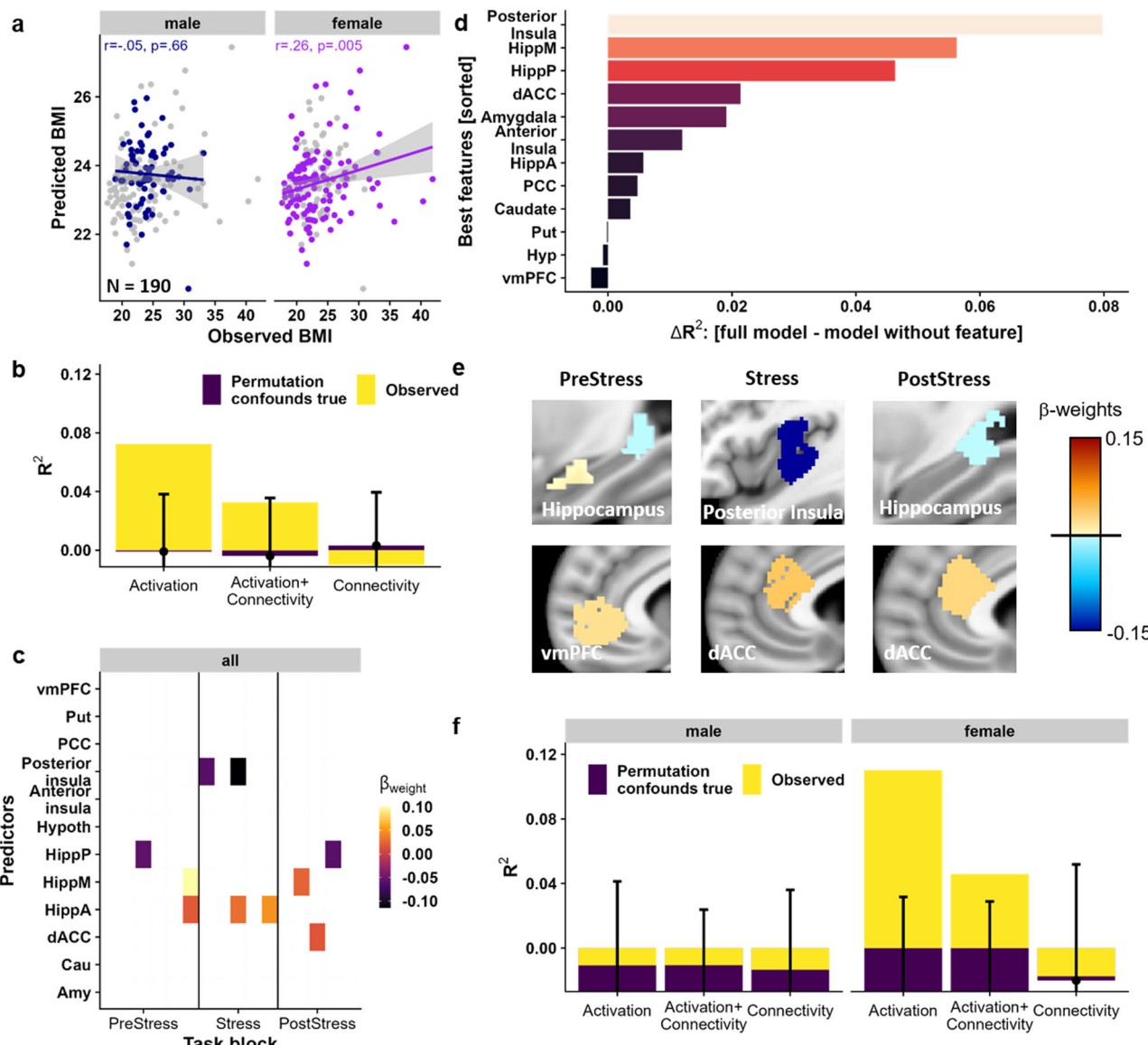

**Fig. 4 Block-wise changes in activation across the task are related to body mass index (BMI) in females. a** An elastic net model based on activation changes during the task predicts BMI. Predicted and observed values of BMI in $n = 190$ participants were significantly correlated across the complete sample ($r(188) = 0.33$, $p_{perm} < 0.001$). This association was driven by $n = 120$ females ($r(118) = 0.26$, $p = 0.005$), but was not seen in $n = 70$ men ($r(68) = -0.05$, $p = 0.66$). Prediction models included covariates (age, sex, diagnosis, pre-task cortisol, and log-transformed average framewise displacement), but for visualization, data is unadjusted. **b** The model based on stress-induced activation trajectories (yellow) predicted BMI beyond a baseline model based on confounding variables. The observed $R^2$ (yellow) from all $n = 190$ participants is higher than the 95% percentiles (errorbars) of the model with true confounds but permuted features (repeated 10,000 times). In contrast, models based on functional connectivity (FC trajectories), or a combination of FC and activation trajectories did not perform better than the baseline model including confounds (i.e., observed $R^2$ within 95% percentile range indicated by the errorbars). Overlapping bars show the average model performance of the observed model (yellow) or the baseline models with permuted features (violet). Error bars depict 95% percentiles derived from permuting (10,000 resamples) the outcome together with the confounding variables to evaluate the contribution of the activation features beyond the confounds. **c** Standardized weights from the prediction model including stress-induced changes in activation. Depicted weights were retained in ≥80% of outer cross-validation folds. **d** Importance of each feature set (i.e., all timepoints of one region) for the prediction of BMI. The $\Delta R^2$ reflects how much predictive accuracy is lost when leaving out all timepoints of the feature. **e** Standardized weights predicting BMI across the complete sample in the model including averaged activations for *PreStress*, *Stress*, and *PostStress*. **f** Activation changes (and changes in activation and FC combined) only predict BMI in $n = 120$ females, but not males, beyond a baseline model including confounds when training separate models. Error bars depict 95% percentiles derived from permuting the outcome together with confounds to evaluate the contribution of features beyond confounds. vmPFC = ventromedial prefrontal cortex, Put = Putamen, PCC = posterior cingulate cortex, Hyp = Hypothalamus, HippP = posterior hippocampus, HippM = medial hippocampus, HippA = anterior hippocampus, dACC = dorsal anterior cingulate cortex, Cau = caudate, Amy = amygdala. Source data are provided in the Supplementary Data 1.

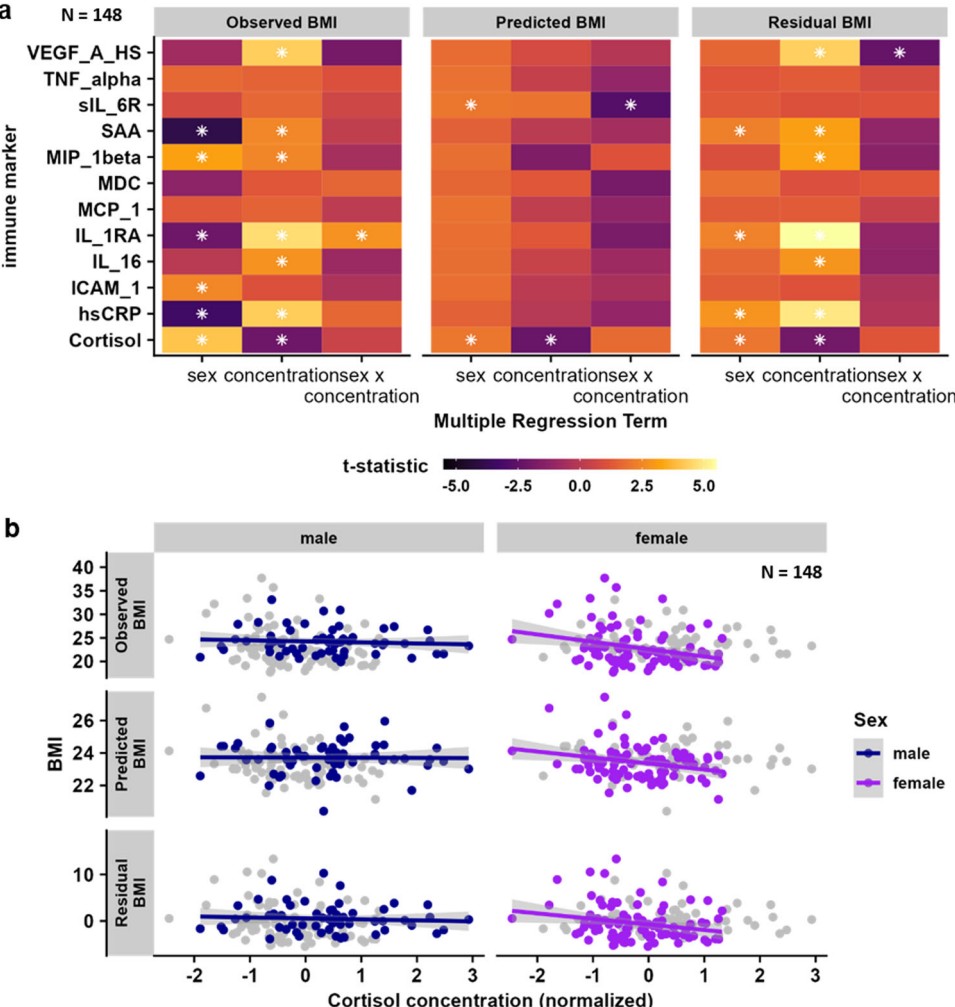

**Fig. 5 Peripheral levels of cytokines do not account for associations with predicted body mass index (BMI) based on stress-induced brain response patterns. a** Multiple regression coefficients for the associations between sex, normalized cytokine concentration, and their interaction with participant's BMI, the BMI predicted by stress-induced brain responses, and the residual BMI in $n = 148$ independent participants. Only baseline (morning) cortisol concentration was related to the observed as well as the predicted BMI. White asterisks indicate significant predictors. All regressions models include age, psychiatric diagnosis, and medication status as additional covariates. **b** Scatterplots for the association between baseline cortisol and BMI (observed, predicted, and residual), split by sex in $n = 89$ females and $n = 59$ males. Associations were numerically stronger and significant in females (rho(83) = −0.27, $p = 0.003$), compared with males (rho(53) = −0.15, $p = 0.22$), but the interaction between sex and cortisol was not significant ($t(142) = 0.46$, $p = 0.64$). Correlation values are partial correlations corrected confounds, but data is shown unadjusted in the scatterplots. Shaded areas show 95% confidence intervals of unadjusted associations. hsCRP = high sensitivity CRP, IL-1RA = interleukin 1 receptor antagonist, VEGF-A = vascular endothelial growth factor A, ICAM-1 = intracellular adhesion molecule 1, MCP-1 = chemokine (C-C motif) ligand 2, MIP-1beta = chemokine (C-C motif) ligand 4, MDC = chemokine (C-C motif) ligand 22, TNF-alpha = tumor necrosis factor alpha, IL-16 = interleukin 16, SAA = serum amyloid A, sIL-6R = soluble IL-6 = receptor. Source data are provided in the Supplementary Data 1.

were uncorrelated with predicted BMI, in contrast to associations with baseline cortisol. To summarize, our results provide initial evidence that acute stress reactivity is more strongly associated with BMI in females compared to males and that these altered responses are more strongly linked to changes in the endocrine, not the immune system.

Across sexes, a higher BMI was related to lower stress-induced activation in the posterior insula, substantia nigra, and posterior hippocampus as well as higher activation in the anterior hippocampus and dACC. Both the hippocampus[47,51,52] and the dACC[53–55] have been repeatedly associated with stress and emotion regulation. An increased weight for hippocampus activity during stress predicting BMI suggests altered regulation of the HPA axis as brain responses of the hippocampus have been associated with stress-induced cortisol responses[47,51]. Likewise, hippocampal activation while viewing aversive pictures was

negatively associated with a baseline cortisol index[56]. Notably, we also observed a negative correlation of baseline cortisol with BMI. Increased dACC activation with a higher BMI might be related to an increased recruitment of the salience network in response to stress[57] and associations during Stress and PostStress might correspond with heighted performance monitoring[58]. Relatedly, lower activation of the posterior insula with a higher BMI during stress in whole-brain and ROI analyses might be related to reduced integration of interoceptive signals and a focus on exteroception[59–64]. The posterior insula has been implicated in the processing of negative social feedback[65] and sex differences[27] as well as associations with negative affectivity[41] have been reported, which has a shared genetic basis with obesity and changes in the immune system[66]. Whole-brain analyses also showed correlations of BMI with stress-induced activation in the substantia nigra, which might correspond with interindividual

differences in stress-induced dopamine signaling[67]. To conclude, our results point to an exaggerated stress response with higher BMI in regions implicated in interoceptive and exteroceptive (salience) processing, including regulation of the HPA axis.

Of note, BMI-associated stress-induced activation trajectories were only observed in females. Accordingly, females also showed stronger associations between BMI and activation in the posterior insula, the subjective stress experience, and baseline cortisol. Since overweight and obesity are more prevalent in women[18,68], our evidence for sex-specific associations are highly relevant. Notably, increased food intake in response to stress is more often reported by women[69–72]. As women also show distinct neural, subjective[29], and endocrine stress responses[23] as well as different associations between hippocampal activity and subjective stress experience[28], it is conceivable that females are more sensitive to altered stress reactivity associated with an increase in BMI. In turn, these sex differences may promote stress-related eating thereby further affecting stress reactivity. This interpretation is supported by negative affective responses to the stressor being related to compensatory food intake[73], which is also associated with altered endocrine stress reactivity[74,75]. Functioning of the HPA axis is affected by sex hormones[20,30] which potentially explains sex- or even menstrual cycle-dependent differences[76]. Notwithstanding, longitudinal studies are necessary to substantiate a potential vicious cycle. Therefore, our results emphasize the role of acute and chronic stress in obesity and overeating particularly in females.

Altered stress-induced brain responses with high BMI might be related to changes in the endocrine and the immune system as well as their interactions[11,33]. Stress-induced cortisol responses were not related to BMI, contrasting earlier studies[31,77,78], although recent evidence has been inconclusive[12–14]. One possible explanation is that not BMI per se, but visceral adiposity is differentially related to HPA-axis functioning[11] since our sample included predominantly overweight females that characteristically have less visceral fat compared to males[79]. In line with previous studies[9,10], a high BMI was associated with lower baseline cortisol levels, especially in females. Critically, lower baseline cortisol levels were also associated with predicted BMI, pointing to shared variance with stress-induced changes in activation that reflect greater adiposity, at least in female participants. Since the HPA axis serves as a negative feedback loop, higher baseline cortisol levels are associated with lower stress-induced endocrine[48,80], but also subjective[48,81], cardiovascular[48], and neural stress responses[48,82]. Accordingly, lower baseline cortisol might reflect an increased potential to react to stress which would be in line with the observed role of the hippocampus. Likewise, exogenous administration of corticosteroids has been associated with changes in brain regions regulating ingestive behavior, which includes the hippocampus, insula, hypothalamus, and the mesocortico-limbic dopamine system[83,84]. In contrast to baseline cortisol, the model-predicted BMI was independent of changes in peripheral cytokines, although a higher BMI was associated with increases in peripheral cytokines as expected[32,33] and this association was stronger in females, comparable to results of neural stress responses. Taken together, our results point to an altered set point of the HPA axis with higher BMI that may mediate altered neural stress reactivity, whereas obesity-related inflammation[35,85,86] potentially has independent effects.

Despite the notable strengths of our study, it has several limitations that call for additional research. First, the sample size is comparatively large for a task-based neuroimaging study, but slightly imbalanced with two thirds of the sample being female, reflecting the differential incidence of mood and anxiety disorders in the population[87,88]. Although the sample covers a broad range of BMI (17.7–41.9 kg²/m) in females, the range was more restricted in males (18.9–33.1 kg²/m). Rerunning analyses with weighted resampling for males to better approximate the female group showed that the associations of BMI with the stress-induced changes in affect changed only slightly (Fig. S6) while sex differences in association with BMI were diminished for cortisol, but not for model-predicted BMI (for details, see SI, Fig. S7). Still, since negative effects of a heightened BMI on the immune system or energy metabolism are conceivably larger in obesity (i.e., BMI > 30 kg/m²)[89–91], a replication of sex-specific effects in a larger and more balanced sample is necessary. Second, we only assessed peripheral inflammation markers, although central inflammation might have distinct mechanisms affecting BMI and stress reactivity[92]. Third, we did not account for effects of the menstrual cycle or the use of hormonal contraception in females, although stress reactivity is strongly affected by the current hormonal state[93,94]. Our study had no exclusion criteria regarding the hormonal state (i.e., contraception or cycle phase) of the participants, but we recorded use of hormonal contraception and the last day of their period when applicable. There were no associations of BMI with use of hormonal contraception or the current cycle day. Hence, longitudinal studies including measurements of sex hormone concentrations are necessary to better understand endocrine modulation[95]. Moreover, other lifestyle factors (e.g., smoking or alcohol consumption) might affect associations between BMI and cytokine levels. Fourth, while sessions all started at the same time to limit circadian effects on stress reactivity, we did not standardize the metabolic state of participants, although glucose levels have been shown to affect stress reactivity[96] and energy metabolism is often altered with higher BMI. Fifth, our results are cross-sectional so we cannot draw conclusions about causal relationships between stress reactivity, overweight and obesity, and associated changes in endocrine or immune systems. Sixth, we only investigated associations with self-reported sex, which was congruent with the sex determined by genotyping in our sample, and a broader scope (e.g., including more diverse samples) could be beneficial.

To summarize, we show that BMI is related to stress-induced activation trajectories of the insula, hippocampus, and dACC in females but not in males. In line with predictive modeling of brain responses, participants with high BMI showed stronger stress-induced changes in negative affect and lower baseline cortisol levels. This may indicate a changed set point of the HPA axis in participants with high BMI, which might contribute to altered stress reactivity, pointing to associations with other stress-related disorders (e.g., depression) that show overlapping genetic factors[66]. To conclude, our results show an important role of altered HPA axis function and acute stress reactivity with higher BMI. Moreover, the observed sex-specific patterns of BMI-associated changes in stress responses in the brain emphasize the need to routinely evaluate such mechanisms separately for males and females[97,98] to improve the treatment of conditions that are linked to altered stress reactivity. If validated, such sex-dependent mechanisms of altered stress reactivity may contribute to sex differences in obesity as well as stress-related eating more broadly.

## Methods

**Participants.** For the analyses reported here, we used a subsample of 192 participants (120 females, $M_{age}$ = 35.0 years ± 12.3 years) from the Biological Classification of Mental Disorders (BeCOME) study (ClinicalTrials.gov: NCT03984084[99]) that completed a psychosocial fMRI stress task and with available BMI data. All participants provided written informed consent at a first inclusion visit, after detailed information was provided by a study physician. The study was approved by the ethics committee of the Ludwig Maximilian University, in Munich, Germany, under the

reference number 350–14 and all ethical regulations relevant to human participant research were followed. The sample covered a broad range of BMI ($Min_{BMI}$ = 17.7, $Max_{BMI}$ = 41.9, $M_{BMI}$ = 23.7, $SD_{BMI}$ = 4.0 kg/m$^2$), particularly in females: (females: range = [17.7–41.9], $M_{BMI}$ = 23.4, $SD_{BMI}$ = 4.6 kg/m$^2$, males: range = [18.9–33.1], $M_{BMI}$ = 24, $SD_{BMI}$ = 2.8 kg/m$^2$). In addition, all participants completed a standardized diagnostic interview[100] and $n$ = 82 (42%) fulfilled the criteria for at least one mood or anxiety disorder (ICD-10 code F3-F4, excluding specific phobias, Table S1) within the last 12 months. Of those, $n$ = 7 reported present medication for their symptoms. Moreover, for 148 participants ($N$ = 89 females), levels of peripheral immune marker and cortisol concentrations were available. To maximize the sample size for the analysis of each stress marker and the predictive modeling[101], we excluded participants with missing or low-quality data for each analysis separately. Specifically, saliva samples of four participants had insufficient biological material. For one participant, the subjective stress experience was not assessed after the stress task. Moreover, data quality of heart rate recordings was not sufficient for peak detection (visual inspection before further analysis) in 25 participants as previously reported[41].

**Experimental procedure**. On the first study day, participants arrived at approximately 8 am. Before the start of the experiments, a blood sample was taken to assess baseline cortisol and cytokines. On the second study day and in the second fMRI session, the stress task (Fig. S1) was included[99]. To assess the cortisol response throughout the task, four saliva samples were taken using Salivettes (Sarstedt AG & Co., Nümbrecht, Germany). Additionally, we assessed the serum cortisol response using blood samples in a subset of participants. The first sample was taken upon arrival (T1) which was followed by the placement of an intravenous catheter (IV) in a subsample ($n$ = 31, 16%) of participants. The second sample (T2) captured a potential cortisol response to the placement of the catheter and was taken approximately 20 min later, directly before entering the scanner. The fMRI session started with an emotional face-matching task (~12 min), followed by a baseline resting-state measurement. Before the start of the stress task (T3), participants rated their current affective state by answering the *Befindensskalierung nach Kategorien und Eigenschaftsworten* (BSKE[102]; SI) via the intercom and a serum sample was taken in the subsample with an IV. We used a psycho-social stress paradigm that was adapted from the Montreal imaging stress task[51]. In this task, participants have to perform arithmetic tasks under time pressure and with negative performance feedback[47,48,99] that is given after each trial and additionally verbally between task blocks. These tasks typically correspond to mild laboratory stressors with 47–65% cortisol responders[103]. It starts with 5 control task blocks (60 s each) interleaved with rest blocks (40 s) of the *PreStress* phase where the arithmetic problems are shown with sufficient time and without negative feedback. This was followed by the *Stress* phase in which the 5 task blocks (again 60 s) are presented with time pressure and negative feedback inducing psycho-social stress. The task ends with a *PostStress* phase that is analogous to *PreStress* to assess stress recovery. The total time of the task is about 25 min. Throughout the fMRI session, we measured heart rate (HR) using photoplethysmography. In the subgroup with additional serum cortisol assessments, two further samples were taken (T4 and T5). After completion of the task, participants rated their current affective state again using the BSKE. Thereafter, another saliva sample was taken (T6) and participants were moved outside of the scanner for a 30 min rest period with an additional serum sample (T7) in the subgroup with an IV. To assess stress-induced

changes in resting-state functional connectivity[104,105], this was followed by another 6-minute resting-state scan. To conclude the session, participants rated their affective state and gave a last saliva sample for cortisol assessment (T8).

**Heart rate measurement: Physiological recording and pre-processing**. As described in Kühnel et al.[48], we measured heart rate using photoplethysmography. Data was acquired with an MR compatible pulse oximeter (Nonin Medical Inc., Plymouth MN, USA) attached to the pulp of the left ring finger. PPG data, sampled at 5 kHz, was amplified using a MR compatible multi-channel BrainVision ExG AUX Box coupled with a BrainVision ExG MR Amplifier (Brain Products GmbH, Gilching, Germany) and recorded with BrainVision Recorder software 1.0. After down-sampling to 100 Hz, RR-intervals were detected using the Physionet Cardiovascular Signal toolbox[106]. Success of detection of beat positions was evaluated by visual inspection. Measurements with insufficient data-quality leading to failed detection of beat positions were excluded ($n$ = 25). Success of beat detection was rated by visually inspecting the detected beat positions. Crucially, the person rating the data was unaware of the patient status of each participant.

**Assessment of subjective stress experience (BSKE scales)**. The BSKE (*Befindlichkeitsskalierung durch Kategorien und Eigenschaftswörter*, Janke, 1994) scales are a short version of the more extensive "*Eigenschaftswörterliste*" (EWL[107], a scale developed to assess the current emotional state across positive and negative dimensions. This reduced scale consists of 15 items (emotions/states) relevant for anxiety that have been previously used to assess stress reactivity[47,48,108] and comparable to the PANAS or state anxiety questionnaire that are also used in stress research[109,110] assess different emotions and feelings that might be affected by stress such as agitation, anxiety, anger, or sensitivity. Participants were asked to rate their current state/feeling ("I feel …") on 6-point scale ranging from 1 ("not at all/gar nicht") to 6 ("very strongly/sehr stark"). We calculated sum scores including the items activity, wakefulness, self-certainty, focus, and relaxed state of mind for positive affect and including the items internal and external agitation, anxiety, sadness, anger, dysphoria, sensitivity as well as three items assessing somatic changes for negative affect.

**Assessment of endocrine and cytokine concentrations**. After collection, all saliva samples were centrifuged and stored at -80° C until further processing. Salivary cortisol concentrations were measured with electro-chemiluminescence-assay (ECLIA) kit (Cobas®, Roche Diagnostics GmbH, Mannheim, Germany). Samples (IDs) were randomized across different batches, but all samples from one participants were processed in the same batch. The detection limit was 1090 pg/mL. The %CV (coefficient of variation) in saliva samples with varying concentrations was between 2.5% and 6.1% for intra-assay variability and between 3.6% und 11.8% for inter-assay variability. Four participants had to be excluded due to insufficient saliva volume at T6.

For a subset of the BeCOME study ($n$ = 198) cytokine assays were measured together with samples from a second in-house study[111]. Blood was collected in Sarstedt plasma tubes at 8.15am in a fasted state and frozen at −80 °C after centrifugation and aliquotating to measure cytokine levels and basal cortisol. The V-PLEX Human Biomarker 54-Plex Kit (Meso Scale Diagnostics, Rockville, USA) was used to measure immune markers in plasma. MSD plates were analyzed on the MSD MESO QucikPlex SQ 120 imager (MSD). Additionally, the following markers were measured with enzyme-linked immunosorbent assay (ELISA):

high-sensitivity C-reactive protein (Tecan Group Ltd., Männedorf, Switzerland, Cat # EU59151), cortisol (Tecan Group Ltd., Männedorf, Switzerland, Cat # RE52061), interleukin 6 (Thermo Fisher Scientific, Waltham, USA, Cat # BMS213HS), interleukin 6 soluble receptor (Thermo Fisher Scientific, Waltham, USA, Cat # BMS214) and interleukin 13 (Thermo Fisher Scientific, Waltham, USA, Cat # BMS231-3). All assays were performed according to the manufacturer's instructions. Cytokines measured with the V-PLEX Biomarker Kit with more than 16% missing values were excluded, resulting in 42 cytokines (Fig. S2) for further analysis. For the markers measured with ELISA, values below the detection limit were set to zero. Likewise, values above the detection limit to the assay were set to the upper limit. All remaining markers were quantile-normalized so values were first ranked and then mapped to the quantiles of a standard normal distribution using custom code. Next, data was batch-corrected for the biobank storage position with linear regression in R version 4.0.2. A linear model was fit for each cytokine to regress out the batch variable and resulting residuals were used for further analysis. Missing values were imputed 100 times with the R package mice 3.13.0, using age, self-reported sex, biobank storage position, Beck Depression Inventory[112], BMI and the study as covariates. For further analysis, the median imputation values were used, as the cytokines included for further analysis (i.e., related to BMI) had at most 2 imputed values (Fig. S3). Of note, baseline plasma cortisol, measured on a different day than the stress task was correlated ($r = 0.37$, $p < 0.001$, Fig. S3) with the first salivary cortisol measurement before the stress task (but after other components of the experimental session).

**fMRI data acquisition and preprocessing**. We acquired MRI data using a 3 T scanner (GE Discovery MR750). The stress task consisted of 755 T2*-weighted echo-planar images (EPI, interleaved acquisition TR = 2 s, TE = 40 ms, 64 × 64 matrix, field of view = 200 × 200 mm², voxel size = 3.5 × 3.5 × 3 mm³). Both resting states consisted of 155 EPIs (TR = 2.5 s, TE = 30 ms, 96 × 96 matrix, field of view = 240 × 240 mm², voxel size = 3.5 × 3.5 × 3 mm³) each. Preprocessing was performed in MATLAB 2018a and SPMv12 as previously reported[41,48]. First, fMRI volumes were corrected for slice-timing. Then, to correct for head motion, fMRI data was realigned to the first image and six movement parameters were derived for later noise correction. For spatial normalization, a high-resolution T2*-weighted image was first segmented using the unified segmentation approach[113]. Derived grey and white matter segments were then used for normalization to the MNI-template by applying DARTEL[113]. Last, data was spatially smoothed with a 6 × 6 × 6 mm³ full-width at half-maximum Gaussian kernel. To perform physiological noise correction, we used aCompCor[114]. To this end, we extracted timeseries of the normalized but unsmoothed functional data of all voxels from white matter and cerebro-spinal fluid segments (probability maps thresholded at $p > 0.90$). We then performed PCA and used the first five components of each segment as physiological noise covariates.

**Concatenation of resting-state and task timeseries**. To assess task-induced functional connectivity changes referenced to a resting-state baseline we concatenated timeseries data from the psycho-social stress task and the two, flanking resting-states. Timeseries were linearly detrended (we did not include a quadratic trend to prevent excluding potential task effects with the same pattern induced by the task structure with non-stress phases flanking the acute stress), despiked, and denoised for each measurement separately so that the average gray scale values of each measurement was 0. To concatenate the task timeseries with the

resting states, we matched the average gray scale values of the flanking resting-states with the average gray scale value of the rest baseline phases (fixation cross) during the *PreStress* condition for each region of interest. To this end, we calculated the average gray scale value (i.e., the measured raw intensity of the fMRI images) after detrending and denoising for each region of interest of the rest baseline phases (fixation cross) during the *PreStress* condition and then subtracted this offset from the complete task timeseries, so that the average intensity value during the rest baseline phases during *PreStress* was 0 and matched the average intensity values of the flanking resting-states.

**Stress response to the psycho-social stress task**
*Endocrine response*. To assess the endocrine response to the stress task, we calculated the change in cortisol concentration between T2 and T6. To account for potential responses of the HPA axis to the placement of the IV[48], we included a dummy-coded nuisance regressor in all analyses that classified participants as pre-task cortisol responders when the concentration at T1 compared to baseline (T0) exceeded 2.5 nmol/l[48,115].

*Autonomous response*. To assess the autonomous stress response[48], we calculated the change in average HR during arithmetic blocks in the *Stress* or *PostStress* phase compared to *PreStress*. After preprocessing raw PPG data and performing beat detection using the Physionet Cardiovascular Signal toolbox[106], we derived the average HR of each task or rest block with the RHRV package[116] for R. This included further preprocessing to exclude implausible interbeat intervals (IBI; exclusion of IBIs <0.3 s or >2.4 s and with excessive deviations from the previous, following, or running average of 50 beats). The threshold for excessive deviations was updated dynamically with the initial threshold set at 13% change from IBI to IBI[117].

*Affective response*. To assess the subjective stress response, we calculated the change in positive (activity, wakefulness, self-certainty, focus, and relaxed state of mind) and negative (internal and external agitation, anxiety, sadness, anger, dysphoria, sensitivity as well as three items assessing somatic changes) sum scores form the respective items directly after the task (T6) and after the 30-min rest interval (T8)[47,48].

*Neural response*. To assess the average neural response to stress across the whole brain, we built a first-level model as previously reported[48]. In this model, three regressors modeled the five arithmetic blocks (60 s each) of the *PreStress*, *Stress* and *PostStress* condition, respectively. To account for motor responses, the model additionally included one regressor modeling individual motor responses. Moreover, the verbal feedback during the *Stress* phase was captured in another regressor. To account for noise components, the models included the six movement parameters, their temporal derivatives, and the aCompCor components (physiological noise components (5 each) from white matter and cerebro-spinal fluid). Data were high-pass filtered with a cut-off of 256 s. To assess the neural stress response and stress recovery we estimated the first-level contrasts of interest, *Stress – PreStress*, and *PostStress – PreStress*, for each participant separately.

**Associations of stress responses with BMI**. Across all stress responses (i.e., subjective, endocrine, autonomous, and neural), we used multiple regression (voxel-wise for neural stress responses) analyses to assess associations with BMI. All analyses additionally included age (associations with BMI, see Table S3), sex (dummy-coded, females as reference), pre-task cortisol, and diagnosis status (fulfilling the criteria for a F3 or F4 diagnosis

within the last 12 months[41]) as confounding variables. Moreover, we explored sex-specific associations of BMI with stress responses by including an interaction term. Analyses of neural stress responses (whole-brain regressions and elastic nets) additionally included average log-transformed framewise displacement[118] as a covariate since BMI has been associated with increased movement during scanning[119].

**Elastic net modeling of BMI based on dynamic neural trajectories.** Next, we evaluated robust associations between the BMI and stress-induced changes in activation and FC within a network defined a priori as being related to stress reactivity and negative affectivity[41]. Negative affectivity is a risk factor for depression and frequently comorbid with obesity especially in women[120,121]. To this end, we used a recently published pipeline that captures dynamic trajectories of activation and FC changes between pre-defined regions of interest (ROI[41]). We then use cross-validated elastic nets to evaluate the generalizability of the associations to unseen data (i.e., held out folds). With these models, we also determine which features of stress-induced changes in brain function best reflect ("predict") BMI in unseen data. This method provides a multivariate model that statistically predicts BMI based on all features ("predicted BMI") and this prediction can subsequently be related to baseline cytokine and cortisol levels for mechanistic inferences. Briefly, average timeseries (unsmoothed) were extracted from the preprocessed task and resting states in 21 ROIs[49] of a stress-related network including the left and right amygdala, hypothalamus, caudate, putamen, anterior, medial, and posterior hippocampus, anterior and posterior insula, and one region for the posterior cingulate, dorsal anterior cingulate, and ventromedial prefrontal cortex. After denoising (detrending, despiking, and residualization with the same regressors as in the whole-brain analyses) and concatenation of the timeseries, hierarchical models for each edge[41,122,123] were used to estimate block-wise changes in activation in 21 ROIs and their FC across, 21*20/2 edges. Each model then included the timeseries of one region (ROI$_1$) as dependent variable and the timeseries of the other (ROI$_2$) as independent variable. Moreover, they included separate regressors for each of the 15 task blocks and their interaction with the predicting ROI timeseries to capture FC changes. Additionally, we accounted for changes in activation corresponding to motor responses or verbal feedback by including the two corresponding convolved regressors from the first-level models. Predictors for interaction terms were centered. In all models, the predictors capturing changes in activation and FC were entered as random effects by participant, so that group-level and regularized individual-level estimates were calculated[124–126]. Next, individual estimates for changes in activation or FC were extracted from each model and aggregated either across regions (activation: combining bilateral regions) or four previously reported subnetworks showing similar stress response trajectories across the task (connectivity[41]): leading to feature sets including 68 (connectivity), 180 (activation), and 248 (activation + connectivity) features. To quantify changes in activation across task blocks, we extracted block-wise estimates from the same linear mixed-effects models we used for the dynamic changes in functional connectivity. Crucially, these models include regressors capturing task-induced changes in activation elicited by task structure (i.e., one regressor for each task block, one regressor for the motor response, and one regressor for the verbal feedback). Since we only estimated the upper triangle of the connectivity matrix, each region of interest was the target region in a different number of models (ranging from 20 to 1). For prediction we used an average across the 210 models based on their anatomical region and combined across all models predicting the same region and subsequently across left and right ROIs leading to 12 × 15 predictors (trajectories across time for all regions) for the prediction of interindividual differences.

Last, we used those block-wise features to predict BMI of held-out folds with elastic net (*lasso*, preset alpha = 0.5, Matlab2020a) and nested 10-fold cross-validation. As in Kühnel et al.[41], we used elastic net since it performs well if the number of features is relatively high and they are correlated[127]. Confounding variables were included in baseline prediction models, and we evaluated the incremental variance explained by fMRI features. Statistical significance was determined using permutation tests (iterations = 10,000; outcome was shuffled with confounders to keep their correlation).

**Contribution of peripheral inflammation.** To determine the contribution of peripheral cytokine levels to changes in stress-induced brain responses associated with a higher BMI, we first selected cytokines that were partially correlated with BMI (see confounds). As we used this step only to select cytokines for further analysis, we did not correct for multiple testing and used an uncorrected $p < 0.05$. Next, we evaluated whether each of the selected cytokines was associated with either the BMI predicted by the brain response when the data was held-out or the residual BMI capturing variance not related to the brain response. If a cytokine is correlated with the predicted BMI, this would indicate that both the stress-induced brain response and cytokine concentration explain shared variance in BMI. In contrast, if it is solely associated with the residual BMI, the cytokine and the stress-induced brain response explain independent variance in BMI. To this end, we used separate multiple regression models for each target cytokine predicting either the predicted BMI or the residual BMI and including the cytokine and the interaction between sex and the cytokine as covariates (as well as the confounds).

**Resampling of male BMI values to match distributions.** To evaluate whether differences between males and females in associations of BMI with cytokines or stress markers can be explained by the difference in the subsamples, we now performed a bootstrapping (1,000 resamples of the data) analysis. In this analysis, observations with very high or low BMIs in males received a higher weight (i.e., where drawn with a higher probability). By adjusting the weights in the male subsample, the mean and standard deviation of males approximated the female distribution. For the relationship of BMI and the negative subjective stress response, we observe only a slight change in the association with BMI in males when mean and standard deviation were matched with the female subgroup (Fig. S5). In contrast, the association of BMI with baseline cortisol in males approached the association in females if distributions became more similar (Fig. S6). However, this was not the case for the association of cortisol with the predicted BMI as the output of the cross-validated elastic net model. This suggests that although the correlations between BMI and cortisol were not sex specific, they are primarily related to differential alterations in the neural stress response in females potentially pointing to different mechanisms.

**Statistics and reproducibility.** We used linear multiple regression models in R to estimate stress effects (i.e., intercept) and associations with BMI, sex, and their interaction across the subjective, cardio-vascular, and cortisol stress response. Likewise, we performed voxel-wise linear multiple regression for the whole-brain analyses. We used partial correlations to determine which cytokines were related to BMI, before again using linear multiple regression models to associate immune markers, sex, and their

interaction with observed BMI and the model predicted BMI. Performance of the elastic net models predicting BMI based on brain responses (i.e., activation and FC) to stress, was evaluated using permutation tests. Here, we permuted the outcome together with baseline covariates (e.g., age, diagnosis status, previous cortisol response, movement) 10,000 times to derive a null distribution of the coefficient of determination ($R^2$) and compared the observed $R^2$. We repeated this procedure for an elastic net model across all 190 individuals and separately for the 120 females and 70 males.

**Reporting summary**. Further information on research design is available in the Nature Portfolio Reporting Summary linked to this article.

## Data availability

Statistical maps showing for the correlations of BMI with stress-induced activation (Stress-PreStress) for the sample and males and females separately are publically available on neurovault (https://neurovault.org/collections/NABGNECT/). A reporting summary for this Article is available as a Supplementary Information file and source data for all figures is available as supplementary data. The raw individual data that support the findings of this study are available from the corresponding author upon reasonable request.

## Code availability

No customized code is necessary to analyze the provided data and MATLAB code used to preprocess the data will be provided upon reasonable request. Predictive modeling was implemented in Matlab 2020a using first linear mixed models (*fitlme*) and elastic net linear models (*lasso*). Statistical analyses (linear models and (partial) correlations) were performed in Rv4.0.2[128]. We used SPM12 for whole-brain fMRI analyses, the voxel threshold was set at $p_{uncorrected} < 0.001$. Clusters were considered significant with a cluster-corrected threshold of $p_{cluster.FWE} < 0.05$. Imaging results were visualized using Mango image processing software (Lancaster, Martinez; www.ric.uthscsa.edu/mango).

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

## Acknowledgements
We thank Anna Hetzel and Ines Eidner for their help with data acquisition, Manfred Uhr and the team of the Max Planck Institute of Psychiatry Biobank for sample processing. N.B.K. received support from the Deutsche Forschungsgemeinschaft (DFG) grants KR 4555/7-1, KR 4555/9-1, and IRTG 2804. V.I.S. has received income from consultations and advisory services for Roche. J.K.-A. was supported by the Brain Behaviour Research Foundation (NARSAD Young Investigator Grant, #28063).

## Author contributions
E.B.B., P.G.S. and the BeCome working group were responsible for the concept and design of the BeCOME study. The BeCome working group was responsible for organization, data acquisition, and data management. M.C. and P.G.S. validated the paradigm and procedure. A.K. and N.B.K. conceived the method and analysis plan, and A.K. performed the data analysis. A.K. wrote the first draft of the manuscript and N.B.K. contributed to the writing. J.H., M.K., and J.K.-A. contributed to the cytokine measures. All authors contributed to the interpretation of findings, provided critical revision of the manuscript for important intellectual content and approved the final version for publication.

## Funding

## Competing interests
The authors declare no competing interests.

## Additional information

**BeCOME working group**

Tanja Brückl[2], Victor I. Spoormaker[2], Angelika Erhardt[2], Norma C. Grandi[2], Julius Ziebula[5], Immanuel G. Elbau[2,5] & Susanne Lucae[5]

