## [Peer Review File · Communications Biology]

Reviewers' comments:

Reviewer #1 (Remarks to the Author):

Kühnel et al. demonstrated that stress-induced changes in negative affect are larger in women with higher BMI, and these stress responses are associated with various brain regions such as the posterior insula and substantia nigra. BMI of women was specifically predicted by their model. On the other hand, they found a correlation of stress reactivity and cortisol (well-known), but not cytokines. These analyses were applied to the datasets from the BeCOME study.

Overall, their analysis results and data themselves are highly reliable. However, based on numerous early reports on obesity, stress, and gender differences, I could not find the outstanding features of this study and what was its significant advances across the biological sciences that are required for publication in Communications Biology.

Major:

(1) The interesting point of this paper is that BMI could be predicted only in women based on brain activity quantified by fMRI. While the association between obesity, stress, and inflammation is a topic of considerable interest, the significance of inflammatory cytokines in this study is not clear in the current manuscript. Even considering the result that there was no correlation between the predicted BMI and the baseline cytokines, the emphasis on inflammation is excessive and the focus of the study is ambiguous.

(2) The main point of study is sex differences in several analyses. However, as they discussed in the Discussion, there are crucial issues in the differences in sampling itself between men and women (such as BMI ranges and sample sizes). If they compared samples with similar indices between men and women, they might be able to find a similar difference from men.

(3) Why was the predicted BMI restricted to the narrow range (22-26: normal weight), although they had more samples with wider BMIs (18-41: underweight to obese)? Could the lack of correlation between the predicted BMI and cytokines be due to such inaccurate BMI prediction?

(4) In Figure 3, it is suggested that in subjects with high BMI, stress-induced inactivation occurred in the substantia nigra and posterior insular cortex (and possibly also in the parietal cortex, although not shown in the figure). In Figure 4, BMI is predicted using network analysis as shown in a previous report, and the brain regions used for the calculation are limited to the same regions as in the previous report, but the brain regions associated with BMI in Figure 3 were not clearly described. The brain regions used in Figure 4 may have been optimized for predicting stress reactivity, but it may be better to incorporate the brain regions observed in Figure 3 in the analysis when predicting BMI. At the very least, there should be some relationship between Figure 3 and Figure 4.

(6) Based on their results, it would be good to further discuss the biological mechanisms (e.g. the roles of brain regions, transmission, or brain-immune interactions) for the difference between men and women in terms of predicting BMI. If there is a gender difference in the relationship between BMI and inflammatory cytokines, it would be a good rationale to focus on inflammatory cytokines.

(Minor)

(1) What is "FC"? (possibly, functional connection?)

(2) In some analyses? What is "b"? If they applied t-test, does it mean t value? In any cases, please clearly describe the number of samples used for analysis or the degree of freedom.

(3) Some figures have several statistics in a graph? Did they truly apply correction for multiple comparisons (e.g. Bonferroni correction)?

(4) Please standardize terminology of brain regions in Figure 3 and 4.

(5) Please provide more detailed explanations for each figure in figures 4 and 5 in the main text.

Reviewer #2 (Remarks to the Author):

In the manuscript Stress-induced brain responses are associated with BMI in women, the authors investigated the relationship between BMI and subjective as well as neural, autonomous, and endocrine responses to an experimental laboratory stressor. This was done on a relatively larger sample for neuroimaging studies (189). The authors investigated these associations further in men and women separately. Overall, increased affective reactivity to stress was associated with BMI, and both increases (precuneus ad parietal lobes) and decreases (posterior insular and substantia nigra) in BOLD activity were seen with higher BMI. Machine learning models were used to see whether changes in functional connectivity can "predict" BMI.

The study is interesting and provides some findings on associations between BMI and stress. While the study is indeed well powered, I have some concerns mostly regarding rationale and analysis. Specifically, the claim of prediction of BMI is unwarranted given the retrospective nature of the study. In my opinion, the core findings of the first statistical models are sufficient, without further need to apply machine learning to a problem that does not necessarily require or warrant it. Below are point-by-point comments for each section.

Introduction:

1. Generally, I am missing background on gender related differences in BMI AND/OR neural stress reactivity. Given that the results heavily lean into gender effects, I think a larger discussion of what has been found previously is warranted. I understand limitations of word counts in this process, but finding a way to incorporate more literature on the topic is important for justifying the rationale.

Methods:

1. The research population description is a bit fuzzy. Some clarifications regarding the final included sample would be good within the text, and not only as a supplementary table.
2. Justification of the reason why participants with incomplete data were included would also be good.
3. Under heading 2.4.4, The authors used a permutation test to determine significance with 1,000 iterations. A 10-k fold cross validation on the 189 participants would likely result in more iterations than used in the permutation test. In my opinion, 1,000 iterations are not enough to ensure sufficient resampling of the data. I would advise the authors to run the model with at least 10,000 iterations. If computational time is an issue, I would suggest testing the results of iterations from 1,000 to 5,000 (in steps of 1,000) and examining the stability of each model. If models are indeed stable with every added iterations than 1,000 would be enough.
4. I find the justification of the elastic net models lacking. What added value does this give us beyond the regression results?

Results:

1. Important demographic information necessary for evaluation of the claims is necessary, such as the number of participants (male and female), the average, standard deviations, and quartile ranges of the BMI. Without this information, it is hard to determine the impact of the results.
2. Page 14, last paragraph, and figure 1. It would be useful to have the exact timepoints in the text and the figures.
3. Overall, given the lack of significant BMI*Sex interactions, I cannot see why the analysis was split on the basis of gender later. A non-significant interaction, followed by the investigation of lower order effects is generally not done, unless prior justification is given. If there was an initial idea of looking at gender effects, this needs to be reflected more in the introduction. If this was a post-hoc decision, this

needs to be explicitly stated and justified, and p-values should be adjusted accordingly to correct for the two separate tests that were done for men and women.

4. Page 17 – last paragraph, last couple of lines, “As for negative affect, the association with BMI was only significant in women”. It is unclear what is meant by this, and what analysis was run.

5. I do not see the added value of the elastic net regressions over the models previously ran.

6. The authors claim predictive models of BMI, however the data was collected retrospectively. No predictions can be made on this basis. If the authors want to make the claim that neural stress responses predict BMI, BMI should at least be tested in the future, otherwise the elastic net models are just offering the same information as the regression models.

Discussion

1. Paragraph 2, page 25. Lower baseline cortisol levels with higher BMI. Could there be a ceiling effect in the lower BMI participants? Specifically, I wonder if there just wasn't as much of an increase in cortisol levels because of higher baseline levels in the low BMI group. The HPA-axis functions with a negative feedback loop, meaning that higher levels of cortisol may act to naturally suppress the further production of cortisol following stress. This is a very important distinction to make, as the exaggerated responses seen may simply come from higher capacity for cortisol output in a group with lower baseline levels.

2. The authors use the term prediction again here, but not attention is given to potential order effects. Perhaps higher stress responsivity promotes stress eating, resulting in higher BMIs. I think more attention needs to be given between the pathways of higher stress reactivity that can lead to increased BMI, instead of only looking at direct correlations between stress and BMI.

General

1. Sex is used consistently throughout the text. However, the graphs and text say Men and Women. Sex in current language refers to the biological distinction (Male/Female), while gender is a spectrum that contains a multitude of identities. I would advise the authors to correct this throughout the manuscript as appropriate. If gender was measured, then it should be noted whether women and men were all cis-gender. If it is indeed sex, then the terms used through-out the manuscript should be changed to male/female.

Reviewer #3 (Remarks to the Author):

The current study investigates the relationship between stress reactivity, increased BMI and brain activity.

Overall, I have very few points of criticism for the manuscript. My biggest concern is that the authors did not control for hormonal status of their female participants. As cycle-dependent fluctuations in sex hormones can affect cortisol levels and possibly interact with the immune system, this could be a strong confounding factor in the analyses. While the authors address this shortly in the discussion, I feel that this possible confound is not discussed in enough detail. I would like the authors to add more information on this as to inform the reader of possible interactions, especially since the effects were only found in women.

The paper is well written. Style and grammar are mostly without errors. Some sentences beginning with introductory clauses are missing a comma after said clause.

On page 8, the sentence beginning with "Throughout the fMRI session": T5 is given in brackets twice. I believe it should be T5 and T6, thus making it T7 in the following sentence.

In total, I can recommend this manuscript for publication with minor revisions.

Reviewer #1 (Remarks to the Author):

Kühnel et al. demonstrated that stress-induced changes in negative affect are larger in women with higher BMI, and these stress responses are associated with various brain regions such as the posterior insula and substantia nigra. BMI of women was specifically predicted by their model. On the other hand, they found a correlation of stress reactivity and cortisol (well-known), but not cytokines. These analyses were applied to the datasets from the BeCOME study.

Overall, their analysis results and data themselves are highly reliable. However, based on numerous early reports on obesity, stress, and gender differences, I could not find the outstanding features of this study and what was its significant advances across the biological sciences that are required for publication in Communications Biology.

We thank the reviewer for their careful evaluation of the manuscript and their helpful suggestions to further improve it. In the revised version of the manuscript, we have now emphasized the rationale for including cytokines in this study. Although the associations between obesity and stress are relatively well understood in rodents and stress-related modulation of eating behavior has received much attention in humans, there is, to our knowledge, no human study investigating stress-induced changes in brain responses in association with BMI. Whether such associations differ between sexes has been largely neglected in mechanistic studies and recent calls have emphasized this problem, particularly in neuroscience (Hodes & Kropp, 2023; Rechlin et al., 2022). Moreover, showing that BMI-related increases in peripheral cytokines are largely independent of altered acute stress responses is crucial to better understand the contributions of different systems implicated in mental and metabolic health.

Major:

(1) The interesting point of this paper is that BMI could be predicted only in women based on brain activity quantified by fMRI. While the association between obesity, stress, and inflammation is a topic of considerable interest, the significance of inflammatory cytokines in this study is not clear in the current manuscript. Even considering the result that there was no correlation between the predicted BMI and the baseline cytokines, the emphasis on inflammation is excessive and the focus of the study is ambiguous.

We thank the reviewer for pointing out that balance between the different components of the manuscript did not reflect the results. Based on the literature, there is substantial evidence on the relationship between the immune system, HPA axis regulation (i.e., stress reactivity), and energy metabolism. Generally, stress responses mediated by the paraventricular nucleus of the hypothalamus prepare the organism to deal with a stressor via the HPA axis and sympathetic nervous system. They include the mobilization of energy stores (Rabasa & Dickson, 2016), but also transient increases in inflammatory cytokines (Rohleder, 2019). In turn, the current state of the energy metabolism (Harrell et al., 2016) and the immune system (Edwards et al., 2010) again affect the stress response. Therefore, it is not trivial that we find no support for the idea that BMI-associated changes in inflammation are also mechanistically linked to BMI-associated changes in stress-induced brain responses (i.e., statistically predicted BMIs based on neural stress responses). Although the literature also highlights that all involved systems show marked sex differences, there has been no human neuroimaging study to date combining measures of inflammation and (neural) stress reactivity that cover a broad range of BMI and symptoms of psychopathology to robustly investigate the associations between all three systems (stress reactivity, BMI, and inflammation) to evaluate sex differences. Therefore, we are convinced that investigating this interaction and reporting both significant and non-significant associations is worthwhile and provides the relevant contribution to

the literature that Communications Biology strives to publish. Nonetheless, we have streamlined the sections on inflammation and included more information on potential sex differences in obesity-related inflammatory processes in the introduction.

p. 5:

Mirroring sex differences in stress reactivity and obesity, the immune system markedly differs between males and females (Klein & Flanagan, 2016). Obesity is strongly associated with increased inflammation in females (Cartier et al., 2009; Thorand et al., 2006) and sex hormones have been proposed to explain the sex dependent interplay of obesity, stress reactivity, and the immune system (Pasquali et al., 2008; Varghese et al., 2017). To conclude, obesity is linked to inflammation in a potentially sex-dependent manner and interactions with the endocrine and the immune system may tune acute stress responses, potentially mediating the effects of obesity on stress.

(2) The main point of study is sex differences in several analyses. However, as they discussed in the Discussion, there are crucial issues in the differences in sampling itself between men and women (such as BMI ranges and sample sizes). If they compared samples with similar indices between men and women, they might be able to find a similar difference from men.

The reviewer raises an important point. Indeed, the distribution of BMI differs between males and females. To evaluate whether differences between males and females in associations of BMI with cytokines or stress markers can be explained by the difference in the subsamples, we now performed a bootstrapping (1,000 resamples of the data) analysis. In this analysis, observations with very high or low BMIs in males received a higher weight (i.e., where drawn with a higher probability). By adjusting the weights in the male subsample, the mean and standard deviation of BMI in males approximated the female distribution. For the relationship of BMI and the negative subjective stress response, we observe only a slight change in the association with BMI in males when mean and standard deviation were matched with the female subgroup (Figure R.1, lower panel).

Figure R.1: Bootstrapped associations of BMI with negative affect after the stress task. In the analysis, data was resampled in males so that the mean (upper right) and standard deviation (upper left) approach the female distribution. For each weighting scheme (x-axis), data was resampled 1,000 times to derive average estimates and 95% confidence intervals. Associations of BMI with stress-induced negative affect in males did only change marginally after adjusting the weights.

In contrast, the association of BMI with baseline cortisol in males approached the association in females if distributions became more similar (Figure R.2). However, this was not the case for the association of cortisol with the predicted BMI as the output of the cross-validated elastic net model. This suggests that although the correlations between BMI and cortisol were not sex specific, they are primarily related to differential alterations in the neural stress response in females potentially pointing to different mechanisms.

Figure R.2: Bootstrapped associations of negative affect after the task with BMI. In the analysis, data was resampled in males so that the mean (upper left) and standard deviation (upper right) gradually approach the female distribution. For each weighting scheme (x-axis), data was resampled 1,000 times to derive average estimates and 95% confidence intervals. Associations of the observed BMI with baseline cortisol became increasingly similar between males and females if the distributions became more similar. In contrast, this was not seen for the correlation with the predicted BMI.

We have now added this context to our statement in the limitations section and expanded on the additional analysis in the supporting information.

p. 29:

Although the sample covers a broad range of BMI (17.7–41.9 kg²/m) in females, the range was more restricted in males (18.9–33.1 kg²/m). Rerunning analyses with weighted resampling for males to better approximate the female group showed that the associations of BMI with the stress-induced changes in affect changed only slightly (Figure S6) whereas sex differences in association with BMI were diminished for cortisol, but not for model-predicted BMI (for details, see SI, Figure S7). Still, since negative effects of an increased BMI on the immune system or energy metabolism are conceivably larger in obesity (i.e., BMI >30 kg/m², Monteiro and Azevedo, 2010; Wisse, 2004; Yudkin, 2007), a replication of sex-specific effects in a larger and more balanced sample is necessary.

(3) Why was the predicted BMI restricted to the narrow range (22-26: normal weight), although they had more samples with wider BMIs (18-41: underweight to obese)? Could the lack of correlation between the predicted BMI and cytokines be due to such inaccurate BMI prediction?

The reviewer makes an excellent observation. The predictive elastic net model, based on the changes in activation, explained about 8% of the variance in BMI. Compared to other predictive models based on similar task-based data, this is a good predictive accuracy (Sui et al., 2020) and clearly significant compared to permuted data. The restricted range simply arises from “conservatism” in the predictive model. In other words, making a prediction close to the mean of the sample is usually a safe bet,

unless there is convincing evidence for a deviation. The restricted range essentially reflects the so-called shrinkage of estimates (Finn et al., 2015). Crucially, this does not strongly affect the relative information contained in the estimates if we turn again to inter-individual differences as this is primarily a form of regularization that pulls extreme observations towards the mean. Still, the predicted BMI specifically captures the variance in BMI robustly explained by differences in stress-induced brain activation patterns. Gaining predicted BMIs that reflect differences in neural response and still recover differences between participants, offers the possibility to determine whether BMI-related altered neural stress reactivity is also related to increased cytokine concentrations even if the BMI range is not completely recovered. This is comparable to an undirected mediation analysis (Cytokines \leftrightarrow stress reactivity \leftrightarrow BMI) without implying causality. We a priori hypothesized that BMI-associated alterations in stress reactivity and peripheral cytokines would be related, as the immune system and the HPA axis are tightly coupled (Rohleder, 2019) and stress-reactivity has been associated with current cytokine levels (Edwards et al., 2010; Kunz-Ebrecht et al., 2003). However, the lack of correlation between the predicted BMI by the model and the cytokines suggests that associations of BMI and altered neural stress reactivity are statistically independent of increased peripheral cytokine levels with a higher BMI. We have now emphasized our reasoning for the analysis as well as the interpretation of the restricted predicted BMI range in relation with the cytokines in the revised manuscript.

p. 21:

The model successfully captured BMI based on activation ($\Delta R^2 = .07$, $p_{\text{perm}} = .0032$, Figure 4A) and including FC did not improve prediction (for predictive accuracies (see Figure 4B, yellow). Of note, the BMI predicted by the elastic net model covered a smaller range than the observed BMI, as predicting values close to the mean is less penalized, leading to shrinkage (Finn et al., 2015) while the relative information between participants is largely unaffected.

(4) In Figure 3, it is suggested that in subjects with high BMI, stress-induced inactivation occurred in the substantia nigra and posterior insular cortex (and possibly also in the parietal cortex, although not shown in the figure). In Figure 4, BMI is predicted using network analysis as shown in a previous report, and the brain regions used for the calculation are limited to the same regions as in the previous report, but the brain regions associated with BMI in Figure 3 were not clearly described. The brain regions used in Figure 4 may have been optimized for predicting stress reactivity, but it may be better to incorporate the brain regions observed in Figure 3 in the analysis when predicting BMI. At the very least, there should be some relationship between Figure 3 and Figure 4.

The reviewer mentions an important point and we thank them for raising it. Indeed, the network of stress-related regions of interest does not include all of the regions revealed by the whole-brain analyses, although the posterior insula (same ROI definition in Figure 3 and 4) is overlapping. Crucially, we defined the brain regions *a priori* since they have been implicated in stress reactivity and also showed a robust association with negative affectivity (Kühnel et al., 2022), a risk factor for mood and anxiety disorders that are comorbid with obesity. To prevent overfitting of any predictive model (Scheinost et al., 2019), a network should be defined *a priori*. Nevertheless, the whole-brain results reported in our study may fuel further work on predictive models in independent samples and we agree with the reviewer that there might be some blind spots with regard to inter-individual differences in energy metabolism. However, such an extension must be evaluated on a new data set to avoid double dipping of the results (Scheinost et al., 2019). To aid future studies extending the network, we shared all unthresholded results maps (correlation with BMI across the whole sample as well as males and females separately) in neurovault. Still, we believe that results from the predefined network highlight the potential of predictive modeling to gain a better understanding of sex-specific alterations in stress-induced brain responses with a higher BMI. Nevertheless, we have

now added the necessity of expanding the 'stress-related' network in future research to capture all potential alterations to the discussion.

p. 27:

Whole-brain analyses also showed correlations of BMI with stress-induced activation in the SN, which might correspond with interindividual differences in stress-induced dopamine signaling (Suridjan et al., 2012).

p. 25:

Additionally, whole-brain analyses revealed stronger stress-induced responses in the substantia nigra and the parietal cortex of participants with higher BMI, suggesting that a more extensive stress-associated network is affected. Notably, associations of BMI with stress-induced changes were not spatially correlated in males and females, pointing to sex-specific associations (see the corresponding maps on neurovault (<https://neurovault.org/collections/NABGNECT/>)).

(6) Based on their results, it would be good to further discuss the biological mechanisms (e.g. the roles of brain regions, transmission, or brain-immune interactions) for the difference between men and women in terms of predicting BMI. If there is a gender difference in the relationship between BMI and inflammatory cytokines, it would be a good rationale to focus on inflammatory cytokines.

We thank the reviewer for this excellent suggestion. Why the stress response in the human brain is mainly associated with BMI in women is an intriguing question for future research, but we can indeed reason based on previous findings, even if comparable designs are scarce to date. In general, sex differences in the immune system and HPA axis have been reported (Kajantie & Phillips, 2006; Klein & Flanagan, 2016). More specifically, sex differences in interaction with BMI have been observed in the response of the HPA axis (Incollingo Rodriguez et al., 2015) and immune system (Varghese et al., 2017). For example, an increased BMI is only associated with increased inflammation in females but not males (Cartier et al., 2009; Thorand et al., 2006). Likewise, increased cortisol reactivity to stress and altered HPA axis functioning is predominantly reported in abdominal obesity that is more common in males compared to females (Incollingo Rodriguez et al., 2015). As a caveat, many studies on eating include only women as they show more stress-related snack intake (e.g., Benson et al., 2009; Epel et al., 2000; Newman et al., 2007; Tomiyama et al., 2011). More broadly, while obesity is more prevalent in females (Ng et al., 2014), obese males seem to be more affected by negative health consequences such as diabetes (Onat et al., 2016; Tramunt et al., 2020). A mechanistic explanation for such differences might be the interaction of the HPA axis and the immune system with sex hormones such as estradiol or testosterone (Pasquali et al., 2008; Varghese et al., 2017). Regarding the role of specific brain regions, differences in stress-induced activation between males and females have been shown (Cohen et al., 2023; Kogler et al., 2015, 2017; Lee et al., 2014). Specifically, evidence from multiple studies suggests that women show stronger stress responses in limbic brain areas (e.g., amygdala), whereas men show stronger responses in regions related to cognitive control such as the dlPFC or IFG and parietal region. Notably, associations between neural stress responses, cortisol (Henze et al., 2021), and subjective stress reactivity (Goldfarb et al., 2019; Kuhn et al., 2023) with for example the hippocampus or frontal regions differed between the sexes, pointing to different neural mechanisms underlying comparable subjective or endocrine responses. However, since differential associations of neural stress reactivity with BMI are not yet well understood, our study seeks to close this knowledge gap.

In our study, we do observe the general associations between immune markers and BMI and across immune markers this association differed between males and females in a multivariate analysis ($p =$

.018). Moreover, peripheral cytokines were unrelated to BMI-related differences in stress-induced brain responses. Nonetheless, BMI-related activation in the hippocampus (i.e., a brain region involved in HPA axis regulation (Herman et al., 2005; Knigge & Hays, 1963)) during stress differed between males and females (BMI * Sex interaction in a voxel-wise analysis) and is only predictive of the BMI in females. Notably, this was mirrored by a correlation of observed and model-predicted BMI with basal cortisol levels again predominantly in females implicating an altered HPA axis set point as potential mechanism linking a higher BMI with altered neural stress reactivity. Therefore, we speculate that the HPA axis dysregulation is more prominent in female vs. male obesity, which may contribute to sex differences in stress-related eating behavior (Grunberg & Straub, 1992; Meule et al., 2018). To better explain the rationale for including immune markers as a candidate mechanism, we have now added more references on sex differences in the link between obesity and the immune system to the introduction and highlighted the potential interpretation of an altered association with the HPA axis in the discussion section.

p. 4:

Mirroring sex differences in stress reactivity and obesity, the immune system markedly differs between males and females (Klein & Flanagan, 2016). Obesity is strongly associated with increased inflammation in females (Cartier et al., 2009; Thorand et al., 2006) and sex hormones have been proposed to explain the sex-dependent interplay of obesity, stress reactivity, and the immune system (Pasquali et al., 2008; Varghese et al., 2017). To conclude, obesity is linked to inflammation in a potentially sex-dependent manner and interactions with the endocrine and the immune system may tune acute stress responses, potentially mediating the effects of obesity on stress.

p. 26:

Accordingly, females also showed stronger associations between BMI and activation in the posterior insula, the subjective stress experience, and baseline cortisol. Since overweight and obesity are more prevalent in women (Garawi et al., 2014; Hedley et al., 2004), our evidence for sex-specific associations are highly relevant. Notably, increased food intake in response to stress is more often reported by women (Adam & Epel, 2007; Grunberg & Straub, 1992; Meule et al., 2018; Zellner et al., 2006). As women also show distinct neural, subjective (Kuhn et al., 2023), and endocrine stress responses (Kajantie & Phillips, 2006) as well as different associations between hippocampal activity and subjective stress experience (Goldfarb et al., 2019), it is conceivable that females are more sensitive to altered stress reactivity associated with an increase in BMI. In turn, these sex differences may promote stress-related eating, further affecting stress reactivity. This interpretation is supported by negative affective responses to the stressor being related to compensatory food intake (Macht & Mueller, 2007), which is also associated with altered endocrine stress reactivity (Herhaus et al., 2020; Newman et al., 2007). Functioning of the HPA axis is also affected by sex hormones (Clark et al., 2022; Pasquali et al., 2008) which potentially explains sex or even menstrual cycle dependent differences (Albert et al., 2015). Notwithstanding, longitudinal studies are necessary to substantiate a potential vicious cycle. Therefore, our results emphasize the role of acute and chronic stress in obesity and overeating particularly in females.

p. 29

Third, we did not account for effects of the menstrual cycle or the use of hormonal contraception in females, although stress reactivity is affected by the current hormonal state (Childs et al., 2010; Kirschbaum et al., 1999). Our study had no exclusion criteria regarding the hormonal state (i.e., contraception or cycle phase) of the female participants, but we

recorded use of hormonal contraception and the last day of their period when applicable. There were no associations of BMI with use of hormonal contraception or the current cycle day. Hence, longitudinal studies including measurements of sex hormone concentrations are necessary to better understand endocrine modulation. Hence, longitudinal studies including measurements of sex hormone concentrations are necessary to better understand endocrine modulation (Schmalenberg, 2022)

(Minor)

(1) What is "FC"? (possibly, functional connection?)

We thank the reviewer for noting the missing definition of the abbreviation as functional connectivity and have corrected it now.

p. 6:

This interdependence is further highlighted as increased inflammation induced by vaccination (Harrison et al., 2009) and in depression (Aruldass et al., 2021) as well as obesity (Park et al., 2020; Syan et al., 2021) has been linked to changes in functional connectivity (FC) in brain networks that are also implicated in stress reactivity (Kühnel et al., 2022).

(2) In some analyses? What is "b"? If they applied t-test, does it mean t value? In any cases, please clearly describe the number of samples used for analysis or the degree of freedom.

We apologize for not communicating clearly that this is the letter *b* according to statistical convention. Since all analyses were multiple regressions, we used *b* in italic to indicate unstandardized regression coefficients (while standardized regression coefficients are typically denoted as β). We now define this at the first occurrence in the manuscript. Moreover, we added t-values with degrees of freedom to all results and additionally include the N of each analysis to the figures directly.

p. 15:

This stress response was indicated by increases in heart rate (during stress: unstandardized estimate (*b*)=6.7, $t(159)=13$, $p<.001$), negative affect (after stress (T6): $b=7.7$, $t(183)=-12.6$, $p<.001$) and decreases in positive affect (after stress (T6): $b=-2.3$, $t(183)=-8.0$, $p<.001$).

(3) Some figures have several statistics in a graph? Did they truly apply correction for multiple comparisons (e.g. Bonferroni correction)?

The reviewer raises an important issue, and we are happy to clarify our approach, including an improved reporting in the revised manuscript. For analyses including multiple outcomes of the same measure (i.e., positive and negative affect at two timepoints to assess the subjective response; changes in heart rate during and after stress to assess the cardiovascular stress response; changes in cortisol directly after the task and after a 30-min break to assess the endocrine response), we first performed a multivariate regression analysis to assess the effects of BMI, sex, and their interaction. Only then, we conducted separate regression analyses for each outcome separately to determine post hoc what was driving the effect for subjective stress responses. To give a complete picture of the data for all stress outcomes, we also performed separate post hoc models for the cardiovascular and cortisol response although there were no multivariate associations. The results of those separate post-hoc test are shown in Figure 2 where we did not correct for multiple testing across those post hoc tests. Likewise, we first performed a multivariate regression across all cytokines to assess whether cytokine concentrations were altered with a higher BMI. For further analyses (i.e., associations with the neural stress response), we only selected a subset of cytokines that was

nominally ($p_{\text{uncorrected}} < .05$) associated with BMI. Within this subset, a multivariate regression furthermore showed that associations with BMI were stronger in females ($p_{\text{MV}} = .019$). In Figure 5 (which depicts separate associations of BMI with cytokine levels), we did not correct for multiple testing as this was only used for feature selection (Finn et al., 2015). For associations of cytokine levels with the model-predicted BMI, a correction was not carried out as even with an uncorrected threshold, we did not observe significant correlations.

p. 17:

Next, we evaluated the effect of sex and BMI on stress reactivity and cytokine levels. On the subjective level, a multivariate regression including changes in negative and positive affect at both timepoints after stress induction revealed that a higher BMI was associated with greater stress-induced changes in affect ($p_{\text{MV}}=.019$).

In line with a link between altered stress reactivity and increased inflammation in overweight and obesity, a higher BMI was associated with increased peripheral cytokine levels ($p_{\text{MV}}=.006$), including increased high sensitivity C-reactive protein (hsCRP), interleukin (IL)-1 receptor antagonist (IL-1RA), tumor necrosis factor (TNF-alpha), IL-16, and soluble IL-6 receptor (sIL-6R) among others (full list of partial correlations with $p < .05$: Figure S2). Cytokine levels were more strongly associated with BMI in females compared to males (BMI×Sex $p_{\text{MV}}=.018$).

(4) Please standardize terminology of brain regions in Figure 3 and 4.

We apologize for the inconsistency in labeling the brain regions across figures and have now harmonized the Posterior insula / InsP to Posterior insula.

(5) Please provide more detailed explanations for each figure in figures 4 and 5 in the main text.

We now provide more detail in the main text and briefly explain all panels as well.

p. 21

The model successfully captured BMI based on activation ($\Delta R^2=.07$, $p_{\text{perm}}=.0032$, Figure 4A) and including FC did not improve prediction (for predictive accuracies (see Figure 4B, observed (yellow) vs. permuted error bars). Of note, the BMI predicted by the elastic net model covered a smaller range than the observed BMI, as predicting values close to the mean is less penalized, leading to shrinkage (Finn et al., 2015) while the relative information between participants is largely unaffected. BMI was predicted by higher activation of the anterior hippocampus, ventromedial prefrontal cortex, and dorsal anterior cingulate cortex (dACC) as well as lower activation of the posterior insula and posterior hippocampus mirroring whole-brain associations (Figure 4C and overlaid on the corresponding ROI 4E, selected features are weights $\neq 0$). Notably, features from the posterior insula, hippocampus and dACC contributed most to the prediction as evaluated by excluding the corresponding feature in the prediction (Figure 4D). Crucially, the elastic net only performed better than chance in females (females: $r=.26$, $p=.005$; males: $r=-.05$, $p=.66$, Sex×BMI: $t(186)=2.7$ $p=.03$, Figure 4A) and re-training the model only in females further improved the accuracy ($\Delta R^2=.11$; $p_{\text{perm}}=.002$, compared to a model including only confounding variables, Figure 4F), although features were similar (Figure S4).

p. 25

Cytokines related with BMI (Figure 5A left panel: multiple regression estimates for the effect of sex, cytokine concentration and their interaction on BMI) were only associated with the

residual BMI and not the predicted BMI (Figure 5A, right panel), suggesting that such differences in inflammation do not account for BMI-associated differences in stress-induced brain responses. In contrast to cytokines, reduced baseline cortisol levels were associated with a higher predicted BMI in a multiple regression model (BMI: $b=-0.31$, $t(142)=-2.30$, $p=.023$; BMI \times Sex: $t(142)=1.73$, $p=.086$, regression estimates Figure 5A, scatterplot Figure 5B).

Reviewer #2 (Remarks to the Author):

In the manuscript Stress-induced brain responses are associated with BMI in women, the authors investigated the relationship between BMI and subjective as well as neural, autonomous, and endocrine responses to an experimental laboratory stressor. This was done on a relatively larger sample for neuroimaging studies (189). The authors investigated these associations further in men and women separately. Overall, increased affective reactivity to stress was associated with BMI, and both increases (precuneus and parietal lobes) and decreases (posterior insular and substantia nigra) in BOLD activity were seen with higher BMI. Machine learning models were used to see whether changes in functional connectivity can “predict” BMI.

The study is interesting and provides some findings on associations between BMI and stress. While the study is indeed well powered, I have some concerns mostly regarding rationale and analysis. Specifically, the claim of prediction of BMI is unwarranted given the retrospective nature of the study. In my opinion, the core findings of the first statistical models are sufficient, without further need to apply machine learning to a problem that does not necessarily require or warrant it. Below are point-by-point comments for each section.

We thank the reviewer for their positive evaluation of the manuscript and their interest in the results. We are positive that integrating the very constructive feedback significantly improved the manuscript so that the rationale for all analyses is now clear.

Introduction:

1. Generally, I am missing background on gender related differences in BMI AND/OR neural stress reactivity. Given that the results heavily lean into gender effects, I think a larger discussion of what has been found previously is warranted. I understand limitations of word counts in this process, but finding a way to incorporate more literature on the topic is important for justifying the rationale.

We thank the reviewer for pointing out the missing background on gender/sex effects in the introduction. On the one hand, sex differences in stress reactivity have been shown across different levels (e.g., subjective, or endocrine). Females show increased subjective responses, whereas males show higher cortisol responses and neural stress reactivity differs between sexes as well (Kogler et al., 2015; Kuhn et al., 2023). On the other hand, studies often report higher incidences of overweight and obesity in females compared to males (Garawi et al., 2014; Hedley et al., 2004) and females report increased food intake after stress more often compared with males (Grunberg & Straub, 1992; Meule et al., 2018). Collectively, this suggests that altered stress reactivity in relation with obesity is more important in females. Similar to behavioral differences, the interaction between the HPA axis and energy metabolism (Bourke et al., 2012; Foss & Dyrstad, 2011; Pasquali et al., 2008) that links stress and food intake may also differ between males and females (Nieuwenhuizen & Rutters, 2008), potentially mediated by sex hormones (Clark et al., 2022). However, many studies investigating the association between stress reactivity and overweight include either only males (to prevent the impact of presumed hormonal fluctuations) or only females. Consequently, we investigated and described sex differences in the link of stress reactivity, including the subjective, endocrine,

cardiovascular, and neural response with BMI. We have added more details regarding potential sex effects to the introduction.

p. 4:

Notably, sex-dependent associations of stress reactivity and obesity are an important mechanism to better understand sex differences in the prevalence of obesity (Hedley et al., 2004; Ng et al., 2014) and its relation to mental (Clark et al., 2022) as well as metabolic disorders (Onat et al., 2016; Tramunt et al., 2020). First, there are sex differences in stress responses as females have shown increased subjective but blunted endocrine responses (Kajantie & Phillips, 2006; Ordaz & Luna, 2012). Likewise, neural stress responses differ between males and females (Cohen et al., 2023; Kogler et al., 2017; Lee et al., 2014), including associations between stress-induced brain responses and subjective stress experiences (Goldfarb et al., 2019; Kuhn et al., 2023). Second, sex hormones regulate endocrine stress response (Pasquali et al., 2008) and energy metabolism (Clark et al., 2022), substantiating potential sex differences in the interplay between stress and BMI. Taken together, there is preliminary evidence for changes in acute stress reactivity in overweight and obesity, but little is known about neural changes or potential sex differences in humans.

p. 5:

Mirroring sex differences in stress reactivity and obesity, the immune system markedly differs between males and females (Klein & Flanagan, 2016). Obesity is strongly associated with increased inflammation in females (Cartier et al., 2009; Thorand et al., 2006) and sex hormones have been proposed to explain the sex dependent interplay of obesity, stress reactivity, and the immune system (Pasquali et al., 2008; Varghese et al., 2017). To conclude, obesity is linked to inflammation in a potentially sex-dependent manner and interactions with the endocrine and the immune system may tune acute stress responses, potentially mediating the effects of obesity on stress.

Methods:

1. The research population description is a bit fuzzy. Some clarifications regarding the final included sample would be good within the text, and not only as a supplementary table.

We apologize for the incomplete information regarding the final sample sizes in the manuscript. We have now added the exact number of exclusions for each of the stress markers together with the reason for exclusion in the main text. Moreover, we added the sample size for each analysis to the corresponding figures.

p. 6:

Specifically, saliva samples of four participants had insufficient biological material. For one participant, the subjective stress experience was not assessed after the stress task. Moreover, data quality of heart rate recordings was not sufficient for peak detection (visual inspection before further analysis) in 25 participants as previously reported (Kühnel et al., 2022)

2. Justification of the reason why participants with incomplete data were included would also be good.

The reviewer raises an excellent question about pairwise versus listwise exclusion. Since the focus of the manuscript are associations of BMI with neural stress reactivity, we maximized the sample size

and statistical power by including all participants with fMRI and anthropometric data to predict the BMI. Maximizing the sample size is important because the reliability of brain–behavior associations depends on the sample size (Grady et al., 2021; Marek et al., 2022) and including only complete data (i.e., listwise exclusion) would reduce the sample size by approximately 25%, even though it is unlikely that data is missing systematically across domains in our study. In such cases, it is strongly recommended to use pairwise exclusion (Kang, 2013). Likewise, we included all available data for each of the analyses regarding the other stress markers (heart rate, cortisol, and subjective stress responses) to maximize the statistical power of each analysis. We have now added the rationale to the description of the sample.

p. 6:

To maximize the sample size for the analysis of each stress marker and the predictive modeling (Grady et al., 2021), we excluded participants with missing or low-quality data for each analysis separately.

3. Under heading 2.4.4, The authors used a permutation test to determine significance with 1,000 iterations. A 10-k fold cross validation on the 189 participants would likely result in more iterations than used in the permutation test. In my opinion, 1,000 iterations are not enough to ensure sufficient resampling of the data. I would advise the authors to run the model with at least 10,000 iterations. If computational time is an issue, I would suggest testing the results of iterations from 1,000 to 5,000 (in steps of 1,000) and examining the stability of each model. If models are indeed stable with every added iterations than 1,000 would be enough.

We thank the reviewer for this suggestion. We have now rerun the permutation test with 10,000 samples. Importantly, we found that the results did not change (Complete sample: $p_{\text{perm}} = .0032$; females only: $p_{\text{perm}} = .0020$), indicating that within this sample, the elastic net model is able to explain significant variance in BMI in cross-validated, held-out samples. We have now updated the results accordingly.

p.14

Statistical significance was determined using permutation tests (iterations=10,000; outcome was shuffled with confounders to keep their correlation).

p. 21:

The model successfully captured BMI based on activation ($\Delta R^2 = .07$, $p_{\text{perm}} = .0032$, Figure 4A-E).

p.22

Re-training and evaluating the model only in females further numerically improved the predictive value ($\Delta R^2 = .11$; $p_{\text{perm}} = .0020$, compared against a baseline model including confounding variables, Figure 4F)

4. I find the justification of the elastic net models lacking. What added value does this give us beyond the regression results?

We apologize for not sufficiently explaining our rationale for the predictive analysis. Although the regression-based analysis of stress-induced activation changes and BMI shows potential associations, the elastic net models provide important additional information. First, cross-validated predictive models guard against over-fitting and overestimation of the association strength by

'predicting' BMI values of unseen data. Cross-validation has increasingly become an important step for establishing brain-behavior associations, as it tests performance in new data, giving a better estimate of out-of-sample performance (model performance estimation). In this way it penalizes very complex models (model selection) overfitting the data and provides estimates of model stability, since the model is retrained across many different data splits (Scheinost et al., 2019; Varoquaux et al., 2017). In brief, the approach, though computationally expensive, helps to suppress false positive findings. Second, predictive models integrate correlated features into one multivariate model instead of using a mass-univariate approach. On the one hand, this enables us to derive 'predicted' BMI based on observed changes in stress responses during fMRI in independent participants (cross-validation). This cross-validation is essential to subsequently assess whether the variance in BMI explained by stress-induced brain responses is associated with peripheral cytokines. On the other hand, it is advantageous to identify features from a candidate set of correlated variables to infer which aspect of stress-induced brain responses is most predictive of the outcome. Third, the elastic net models enable us to also investigate the associations of BMI with changes in functional connectivity in a predefined network that we have shown to recover different stress phases and predict individual differences in negative affectivity (Kühnel et al., 2022).

p. 12 (Methods)

Next, we evaluated robust associations between BMI and stress-induced changes in activation and functional connectivity (FC) within an a priori defined network related to stress reactivity and negative affectivity (Kühnel et al., 2022). Negative affectivity is a risk factor for depression and frequently comorbid with obesity, especially in women (de Wit et al., 2010; Preiss et al., 2013). To this end, we used a recently published pipeline that captures dynamic trajectories of activation and FC changes between regions of interest (ROI, (Kühnel et al., 2022)). We then use cross-validated elastic nets to evaluate the generalizability of the associations to unseen data (i.e., held out folds). In that way, we also determine which features of stress-induced changes in brain function best reflect ("predict") BMI in unseen data. This method provides a multivariate model that statistically predicts BMI based on all features ("predicted BMI"), which can be related to baseline cytokine and cortisol levels for mechanistic inferences.

p. 21 (Results)

To associate stress-induced changes in brain responses with BMI and derive individual predictions for unseen data, we used cross-validated predictive modelling with elastic nets. This prediction captures the variance explained by spatio-temporal profiles of stress-induced changes in brain activation and FC, which we have previously shown to recover negative affectivity beyond conventional analyses (Kühnel et al., 2022).

Results:

1. Important demographic information necessary for evaluation of the claims is necessary, such as the number of participants (male and female), the average, standard deviations, and quartile ranges of the BMI. Without this information, it is hard to determine the impact of the results.

We agree with the reviewer that this information is essential and apologize for not reporting the BMI characteristics of sample in detail in the previous version of the manuscript. The sample includes approximately two thirds females (n=120, in line with the distribution of mental disorders in the population). The BMI ranges from 17.7 to 41.9 kg/m² across the whole sample (females: 17.7 – 41.9

kg/m²; males: 18.9 – 33.1 kg/m²) with an average BMI of 23.4 ± 4.0 kg/m² (males: 24 ± 2.8 kg/m²; females: 23.5 ± 4.6 kg/m²). We have now added this information to the sample description.

p. 6

The sample covered a broad range of BMI (Min_{BMI} = 17.7, Max_{BMI} = 41.9, M_{BMI}=23.7, SD_{BMI}=4.0 kg/m²), particularly in females: (females: range = [17.7-41.9], M_{BMI}=23.4, SD_{BMI}=4.6 kg/m², males: range = [18.9-33.1], M_{BMI}=24, SD_{BMI}=2.8 kg/m²).

2. Page 14, last paragraph, and figure 1. It would be useful to have the exact timepoints in the text and the figures.

We thank the reviewer for pointing out the lack of information in the text. We have now added the exact timepoint each of the reported tests refers to in the text as well.

p.15

As reported in a previous publication (Kühnel et al., 2022), the task elicited a robust multi-level stress response (Figure 1). as This stress response was indicated by an increase in heart rate (during stress: unstandardized estimate (b) = 6.7, t(159)=13 ,p < .001), negative affect (after stress (T6): b = 7.7, t(183)=-12.6 ,p < .001) and a decrease in positive affect (after stress (T6): b = -2.23, t(183)=-8.0, p < .001). Cortisol increased in response to stress (T6 after stress, ~ 16 min after stress onset) in participants not showing an increase in cortisol to the blood drawing procedure (b = 0.4, t(178)=2.6 ,p = .011). After stress, heart rate recovered but not to baseline levels (after stress during recovery: b = 0.8788, t(159)=-2.1 ,p = .034). Moreover, after a thirty30-minute break, negative affect (T8: b = -1.0, t(183)=-4.7 .p = .014), but not positive affect (T8: b = -1.3, t(183)=-2.5 ,p < .001), had recovered.

3. Overall, given the lack of significant BMI*Sex interactions, I cannot see why the analysis was split on the basis of gender later. A non-significant interaction, followed by the investigation of lower order effects is generally not done, unless prior justification is given. If there was an initial idea of looking at gender effects, this needs to be reflected more in the introduction. If this was a post-hoc decision, this needs to be explicitly stated and justified, and p-values should be adjusted accordingly to correct for the two separate tests that were done for men and women.

We thank the reviewer for pointing out the lack of justification in the previous version of the manuscript. Indeed, our idea was to investigate sex interactions in light of recent calls for sex/gender-specific approaches in medicine and neuroscience (Hodes & Kropp, 2023; Rechlin et al., 2022). Such approaches are an important part of the newly established International Research Training Group “Women’s Mental Health Across the Reproductive Years” that we are part of as well. For example, we have recently published work on sex differences in effort allocation as well as an invited commentary for Nature Metabolism on cycle-dependent changes in insulin sensitivity that might explain sex differences in behavior that is dependent on energy metabolism, including mood and stress reactivity. In addition to the theoretical rationale for separate analyses, we have also restructured the results sections to better align the reporting of the findings with the sequence of analyses that we conducted. Briefly, we observed significant interactions in several analyses early on in the process (e.g., association of immune markers and BMI, prediction of BMI based on stress responses) and a striking dissimilarity in the correlations of stress-induced changes in brain responses for males and females. We report these observations now in more detail and a more intuitive sequence in the revised version of the manuscript. In addition, we also ran additional cross-predictive analyses (i.e., a model trained on women to predict responses in men) to show that sex-

specific models are necessary to improve our understanding of the neural processes, even if subjective stress responses are more similar between the sexes.

p.17:

Next, we evaluated the effect of sex and BMI on stress reactivity and cytokine levels. On the subjective level, a multivariate regression including changes in negative and positive affect at both timepoints after stress induction revealed that a higher BMI was associated with greater stress-induced changes in affect ($p_{MV}=.019$). Specifically, a higher BMI was related to more negative affect after the task ($b=1.48$, $t(182)=2.00$, $p=.047$, $N_{females}=120$, Figure 2) and after the 30-min rest period ($b=1.18$, $t(182)=2.35$, $p=.019$), relative to the baseline. This association was significant in females at the later time point (T6: $b=1.7$, $t(115)=1.84$, $p=.069$; T8: $b=1.3$, $t(115)=2.28$, $p=.025$) but not in men (T6: $b=-1.1$, $t(66)=-1.20$, $p=.24$; T8: $b=0.6$, $t(66)=0.76$, $p=.44$; Figure 2B), but the interaction between sex and BMI did not reach significance (T6: $t(182)=-1.68$, $p=.095$; T8: $t(182)=-0.44$, $p=.66$). In contrast, higher BMI was not associated with stress-induced changes in heart rate, or cortisol concentrations ($ps>.10$, Figure 2A, Table S4) and subjective, cardiovascular, and endocrine stress responses did not differ between males and females ($ps>.15$; Figure 2A, Table S2)

In line with a link between altered stress reactivity and increased inflammation in overweight and obesity, a higher BMI was associated with increased peripheral cytokine levels ($p_{MV}=.006$), including increased high sensitivity C-reactive protein (hsCRP), interleukin (IL)-1 receptor antagonist (IL-1RA), tumor necrosis factor (TNF-alpha), IL-16, and soluble IL-6 receptor (sIL-6R) among others (full list of partial correlations with $p < .05$: Figure S2). Cytokine levels were more strongly associated with BMI in females compared to males (BMI×Sex $p_{MV}=.018$). In contrast, baseline cortisol (measured in plasma samples at the beginning of an independent session ~8 am) was lower in participants with high BMI ($\rho = -.17$, $p = .015$) and did not differ between sexes (BMI×Sex $t(142)=0.46$, $p=.64$, $\rho_{female}=-.27$, $p=.003$; $\rho_{male}=-.15$, $p=.22$).

p. 19:

Similar to negative affect, higher BMI was associated with stronger stress-induced decreases of BOLD responses in the posterior insula (L: $p_{FWE} < .001$, $k = 293$, R: $p_{FWE} = .041$, $k = 143$) and a midbrain cluster including the substantia nigra ($p_{FWE} < .001$, $k = 390$) as well as increased BOLD responses in the precuneus/superior parietal lobe ($p_{FWE} = .025$, $k = 145$, Figure 3, for unthresholded maps, see <https://neurovault.org/collections/NABGNECT/>). Notably, within the pre-defined stress-related network, females showed a more negative correlation with BMI in the hippocampus (Sex×BMI: $t_{max}=4.26$, $p_{SVC.Hippocampus}=.008$; $p_{SVC.StressNetwork}=.088$). Moreover, BMI-associated changes in stress responses across ROIs (Shen atlas) calculated separately for males and females were not spatially correlated ($r = .05$, $p = .38$, Figure S5). To evaluate potential sex effects, we conducted post hoc regression analyses (including BMI×Sex interactions) as well as separately for males and females on average beta values extracted from ROIs ((Shen et al., 2013) overlapping the posterior insula and substantia nigra. While the interaction of sex and BMI did not reach significance (posterior insula R: $p=.071$, SN: $p=.36$), associations were numerically higher in females (Figure 3B, for details see SI).

p.22

Crucially, the elastic net only performed better than chance in females (females: $r=.26$, $p=.005$; males: $r=-.05$, $p=.66$, Sex×BMI: $t(186)=2.7$ $p=.03$, Figure 4A) and re-training the model only in females further improved the accuracy ($\Delta R^2=.11$; $p_{perm}=.002$, compared to a model including confounding variables, Figure 4F), although features were similar (Figure S4). Moreover, using

models trained on females to predict BMI in males was not successful (and vice versa), indicating that neural stress responses reflective of BMI differ between sexes.

4. Page 17 – last paragraph, last couple of lines, “As for negative affect, the association with BMI was only significant in women”. It is unclear what is meant by this, and what analysis was run.

We apologize for not explaining the performed analyses more clearly here. Analogous to subjective, endocrine, and autonomous stress responses and their associations with BMI, we performed regression analyses across the whole sample, as well as separately for males and females. We have now explained this more clearly in the text.

p. 18

To evaluate potential sex effects, we conducted post hoc regression analyses (including BMI*Sex interactions) as well as separately for males and females on average beta values extracted from ROIs (Shen et al., 2013) overlapping the posterior insula and substantia nigra.

5. I do not see the added value of the elastic net regressions over the models previously ran.

We thank the reviewer for pointing out that we need to communicate the added value of the elastic net models more clearly (see also our response to comment 4; Methods of reviewer 2). The cross-validated elastic net models have several crucial advantages compared to the conventional regression analyses that we believe provide additional insights, specifically concerning the robustness of the observed associations. Briefly, the elastic net models provide multivariate models to statistically predict BMI in held-out folds (i.e., cross-validation) based on all fMRI features combined. The separation into training and test data guards against overfitting, specifically if an independently defined set of regions of interest is used (as defined a priori in Kühnel et al., 2022, *Biol Psychiatry* already). Moreover, univariate associations are not independent because brain responses are correlated across the brain, so it is difficult to gauge the overall predictive potential solely based on mass-univariate associations. Do additional regions add incremental information or are they essentially redundant? This question is important for mechanistic insight as well. Lastly, the predicted BMI is then used in further analyses to determine whether BMI-associated changes in inflammation are also related to BMI-associated changes in brain responses. In addition to the changes in the methods section mentioned before, we further elaborated on the rationale in the results section.

p.6 (Introduction)

To this end, we first investigated stress-induced changes in brain responses (i.e., activation changes). Moreover, obesity has been robustly related to alterations in FC (Farruggia et al., 2020). We have previously related stress-induced dynamic FC trajectories within a putative stress network with negative affectivity, a risk factor for mood and anxiety disorders that are often comorbid with obesity (Kühnel et al., 2022). Therefore, we derived dynamic FC and activation trajectories and used cross-validated elastic nets to evaluate robust associations of these imaging features with interindividual differences in BMI.

p. 12 (Methods)

Next, we evaluated robust associations between BMI and stress-induced changes in activation and functional connectivity (FC) within an a priori defined network related to stress reactivity and negative affectivity (Kühnel et al., 2022). Negative affectivity is a risk factor for depression and frequently comorbid with obesity, especially in women (de Wit et al., 2010; Preiss et al., 2013). To this end, we used a recently published pipeline that captures

dynamic trajectories of activation and FC changes between regions of interest (ROI, (Kühnel et al., 2022)). We then use cross-validated elastic nets to evaluate the generalizability of the associations to unseen data (i.e., held out folds). With these models, we also determine which features of stress-induced changes in brain function best reflect (“predict”) BMI in unseen data. This method provides a multivariate model that statistically predicts BMI based on all features (“predicted BMI”), which can be related to baseline cytokine and cortisol levels for mechanistic inferences.

p. 21 (Results)

To associate stress-induced changes in brain responses with BMI and derive individual predictions for unseen data, we used cross-validated predictive modelling with elastic nets. This prediction captures the variance explained by spatio-temporal profiles of stress-induced changes in brain activation and FC, which we have previously shown to recover negative affectivity beyond conventional analyses (Kühnel et al., 2022).

6. The authors claim predictive models of BMI, however the data was collected retrospectively. No predictions can be made on this basis. If the authors want to make the claim that neural stress responses predict BMI, BMI should at least be tested in the future, otherwise the elastic net models are just offering the same information as the regression models.

We agree with the reviewers’ concerns regarding the use of the term “predict” in a cross-sectional study. Here, we have been using “predict” strictly in a statistical sense (as in: the cross-validated model predicts), not in the sense of predicting future outcomes in longitudinal studies. Although the elastic net model solely uses cross-sectional data, the nested cross-validation essentially predicts BMI in unseen individuals based on a subset of the observed data. Since such cross-validation guards against overfitting compared to a conventional regression analysis, such models are regarded as more robust for potential generalization to new data set (Scheinost et al., 2019). Another advantage of the elastic net model is the inclusion of correlated predictors, providing a multivariate set of features including selection of the most “predictive” features. To avoid confusion about the kind of prediction we refer to, we have exchanged “predict” for phrases such as cross-validated association or correlation throughout the manuscript whenever we were not specifically referring to the BMI values predicted by the model for each participant.

Discussion

1. Paragraph 2, page 25. Lower baseline cortisol levels with higher BMI. Could there be a ceiling effect in the lower BMI participants? Specifically, I wonder if there just wasn’t as much of an increase in cortisol levels because of higher baseline levels in the low BMI group. The HPA-axis functions with a negative feedback loop, meaning that higher levels of cortisol may act to naturally suppress the further production of cortisol following stress. This is a very important distinction to make, as the exaggerated responses seen may simply come from higher capacity for cortisol output in a group with lower baseline levels.

We agree this is an important point when interpreting the results and thank the reviewer for raising it. Indeed, baseline levels of cortisol are an important factor determining cortisol reactivity to stress (Dickerson & Kemeny, 2004), influencing reactivity across other dimensions such as the cardiovascular system (Gossett et al., 2018; Kühnel et al., 2020), subjective (Kühnel et al., 2020) and neural stress responses (Kühnel et al., 2020; Zschucke et al., 2015). Although imaging stress tasks have been shown to robustly induce cortisol responses, those are often smaller compared to stress induction outside of the MRI. One potential explanation is the response already elicited purely by the MRI environment, leading to a ceiling effect (Gossett et al., 2018). Therefore, one might speculate

that the observed lower baseline cortisol is related to higher stress reactivity in participants with a higher BMI. We have now discussed this pathway in more detail in the manuscript.

p. 27:

Critically, lower baseline cortisol levels were also associated with predicted BMI, pointing to shared variance with stress-induced changes in activation that reflect greater adiposity, at least in female participants. Since the HPA axis serves as a negative feedback loop, higher baseline cortisol levels are associated with lower stress-induced endocrine^{53,106}, but also subjective^{53,107}, cardiovascular⁵³, and neural stress responses^{53,108}. Accordingly, lower baseline cortisol might reflect an increased potential to react to stress which would be in line with the observed role of the hippocampus.

2. The authors use the term prediction again here, but not attention is given to potential order effects. Perhaps higher stress responsivity promotes stress eating, resulting in higher BMIs. I think more attention needs to be given between the pathways of higher stress reactivity that can lead to increased BMI, instead of only looking at direct correlations between stress and BMI.

The reviewer raises an excellent point that we have also discussed repeatedly in preparing the submission. Our 'predictions' are purely statistical and reflect the explained variance in unseen data. Due to the cross-sectional design, we agree that the study cannot determine whether higher stress responsivity promotes stress eating and consequently a higher BMI or, vice versa, a higher BMI alters stress reactivity. In addition to exchanging "predict" for less suggestive phrases throughout the discussion, we also expanded the paragraph speculating how altered stress reactivity and overeating might be related in a bidirectional 'vicious cycle'.

p.29:

Of note, BMI-associated stress-induced activation trajectories were only observed in females. Accordingly, females also showed stronger associations between BMI and activation in the posterior insula, the subjective stress experience, and baseline cortisol. Since overweight and obesity are more prevalent in women^{18,94}, our evidence for sex-specific associations are highly relevant. Notably, increased food intake in response to stress is more often reported by women⁹⁵⁻⁹⁸. As women also show distinct neural, subjective²⁹, and endocrine stress responses²³ as well as different associations between hippocampal activity and subjective stress experience²⁸, it is conceivable that females are more sensitive to altered stress reactivity associated with an increase in BMI. In turn, these sex differences may promote stress-related eating further affecting stress reactivity. This interpretation is supported by negative affective responses to the stressor being related to compensatory food intake⁹⁹, which is also associated with altered endocrine stress reactivity^{100,101}. Functioning of the HPA axis is affected by sex hormones^{20,30} which potentially explains sex- or even menstrual cycle-dependent differences¹⁰². Notwithstanding, longitudinal studies are necessary to substantiate a potential vicious cycle. Therefore, our results emphasize the role of acute and chronic stress in obesity and overeating particularly in females.

General

1. Sex is used consistently throughout the text. However, the graphs and text say Men and Women. Sex in current language refers to the biological distinction (Male/Female), while gender is a spectrum that contains a multitude of identities. I would advise the authors to correct this throughout the

manuscript as appropriate. If gender was measured, then it should be noted whether women and men were all cis-gender. If it is indeed sex, then the terms used through-out the manuscript should be changed to male/female.

We thank the reviewer for raising this important point and now use terms referring to biological sex (self-reported and confirmed with genotyping) throughout the text and figures.

Reviewer #3 (Remarks to the Author):

The current study investigates the relationship between stress reactivity, increased BMI and brain activity. Overall, I have very few points of criticism for the manuscript.

We thank the reviewer for their positive evaluation of the manuscript. We are happy to correct the mistakes and make the suggested changes to further improve the manuscript.

1) My biggest concern is that the authors did not control for hormonal status of their female participants. As cycle-dependent fluctuations in sex hormones can affect cortisol levels and possibly interact with the immune system, this could be a strong confounding factor in the analyses. While the authors address this shortly in the discussion, I feel that this possible confound is not discussed in enough detail. I would like the authors to add more information on this as to inform the reader of possible interactions, especially since the effects were only found in women.

The reviewer raises an important point. In addition to evidence showing differences in stress reactivity and cortisol between males and females (Kajantie & Phillips, 2006; Kuhn et al., 2023), there is also evidence that (cortisol) stress reactivity depends on the menstrual cycle phase, sex hormone concentrations, or the use of hormonal contraceptives (Kirschbaum et al., 1999; Stephens et al., 2016). Importantly, sex hormones may also contribute to the difference between males and females in the relationship of BMI and stress reactivity (Bourke et al., 2012) as well as of BMI and the immune system (Varghese et al., 2017). In the current study, we only asked participants about their use of contraceptives and the date of their last menstruation. Briefly, of the 120 females, 90 were naturally cycling, 20 used contraceptives of any kind and 10 were already in menopause. The BMI did not significantly differ between these groups ($ps > .64$). Of the 90 naturally cycling women, 60 reported the date of their last menstruation as well as 10 participants using contraceptives. There was no linear or quadratic association of the number of days since the last period with BMI ($ps > .60$). However, to rigorously investigate effects across the cycle or different contraceptives, study designs optimized for this question (i.e., longitudinal to assess within-person effects across the cycle or cross-sectional studies with more participants, (Schmalenberger et al., 2021)) would be necessary. Moreover, a direct assessment of sex hormones is necessary to confirm the exact phase each participant is in. In fact, we are currently preparing more suitable studies to address these pressing questions. Therefore, we now added a more in-depth discussion of the potential influences of cycle phase on our results to the limitations section.

p. 29:

Third, we did not account for effects of the menstrual cycle or the use of hormonal contraception in females, although stress reactivity is strongly affected by the current hormonal state (Childs et al., 2010; Kirschbaum et al., 1999). Our study had no exclusion criteria regarding the hormonal state (i.e., contraception or cycle phase) of the female participants, but we recorded use of hormonal contraception and the last day of their period when applicable. There were no associations of BMI with use of hormonal contraception or the current cycle day. Hence, longitudinal studies including measurements of sex hormone concentrations are necessary to better understand endocrine modulation. Hence,

longitudinal studies including measurements of sex hormone concentrations are necessary to better understand endocrine modulation (Schmalenberger et al., 2021).

2) The paper is well written. Style and grammar are mostly without errors. Some sentences beginning with introductory clauses are missing a comma after said clause.

We thank the reviewer for their careful reading of the manuscript and apologize for the omissions. We have edited the paper again for style and grammar.

3) On page 8, the sentence beginning with "Throughout the fMRI session": T5 is given in brackets twice. I believe it should be T5 and T6, thus making it T7 in the following sentence.

We thank the reviewer for bringing this error to our attention. Indeed, the sampling points for the blood serum samples throughout the MRI session are T4 and T5. Consequently, the first saliva sample after the task is at T6. Since another serum, but not another saliva sample is taken during the rest period (T7), the last saliva sample taken at the end of the experiment is T8. We have now added all time points to the experimental procedure in the main text so that it is harmonized with the detailed figure in the supporting information as well as our previous publications (Kühnel et al., 2020, 2022).

p. 7-8

Throughout the fMRI session, we measured heart rate (HR) using photoplethysmography (SI). In the subgroup with additional serum cortisol assessments, two further samples were taken (T4 and T5).

Thereafter, another saliva sample was taken (T6) and participants were moved outside of the scanner for a 30min rest period with an additional serum sample (T7) in the subgroup with an IV.

In total, I can recommend this manuscript for publication with minor revisions.

We thank the reviewer for recommending our manuscript for publication and appreciated receiving the actionable suggestions to improve the submission.

References

- Adam, T. C., & Epel, E. S. (2007). Stress, eating and the reward system. *Physiology & Behavior*, *91*(4), 449–458. <https://doi.org/10.1016/j.physbeh.2007.04.011>
- Albert, K., Pruessner, J., & Newhouse, P. (2015). Estradiol Levels Modulate Brain Activity and Negative Responses to Psychosocial Stress across the Menstrual Cycle. *Psychoneuroendocrinology*, *59*, 14–24. <https://doi.org/10.1016/j.psyneuen.2015.04.022>
- Benson, S., Arck, P. C., Tan, S., Mann, K., Hahn, S., Janssen, O. E., Schedlowski, M., & Elsenbruch, S. (2009). Effects of obesity on neuroendocrine, cardiovascular, and immune cell responses to acute psychosocial stress in premenopausal women. *Psychoneuroendocrinology*, *34*(2), 181–189. <https://doi.org/10.1016/j.psyneuen.2008.08.019>
- Bourke, C. H., Harrell, C. S., & Neigh, G. N. (2012). Stress-induced sex differences: Adaptations mediated by the glucocorticoid receptor. *Hormones and Behavior*, *62*(3), 210–218. <https://doi.org/10.1016/j.yhbeh.2012.02.024>
- Cartier, A., Côté, M., Lemieux, I., Pérusse, L., Tremblay, A., Bouchard, C., & Després, J.-P. (2009). Sex differences in inflammatory markers: What is the contribution of visceral adiposity? *The American Journal of Clinical Nutrition*, *89*(5), 1307–1314. <https://doi.org/10.3945/ajcn.2008.27030>
- Chen, X., Gianferante, D., Hanlin, L., Fiksdal, A., Breines, J. G., Thoma, M. V., & Rohleder, N. (2017). HPA-axis and inflammatory reactivity to acute stress is related with basal HPA-axis activity. *Psychoneuroendocrinology*, *78*, 168–176. <https://doi.org/10.1016/j.psyneuen.2017.01.035>

- Childs, E., Dlugos, A., & Wit, H. D. (2010). Cardiovascular, hormonal, and emotional responses to the TSST in relation to sex and menstrual cycle phase. *Psychophysiology*, *47*(3), 550–559. <https://doi.org/10.1111/j.1469-8986.2009.00961.x>
- Clark, T. D., Reichelt, A. C., Ghosh-Swaby, O., Simpson, S. J., & Crean, A. J. (2022). Nutrition, anxiety and hormones. Why sex differences matter in the link between obesity and behavior. *Physiology & Behavior*, *247*, 113713. <https://doi.org/10.1016/j.physbeh.2022.113713>
- Cohen, J. E., Holsen, L. M., Ironside, M., Moser, A. D., Duda, J. M., Null, K. E., Perlo, S., Richards, C. E., Nascimento, N. F., Du, F., Zhou, C., Misra, M., Pizzagalli, D. A., & Goldstein, J. M. (2023). Neural Response to Stress Differs by Sex in Young Adulthood. *Psychiatry Research: Neuroimaging*, 111646. <https://doi.org/10.1016/j.psychresns.2023.111646>
- de Wit, L., Luppino, F., van Straten, A., Penninx, B., Zitman, F., & Cuijpers, P. (2010). Depression and obesity: A meta-analysis of community-based studies. *Psychiatry Research*, *178*(2), 230–235. <https://doi.org/10.1016/j.psychres.2009.04.015>
- Dickerson, S. S., & Kemeny, M. E. (2004). Acute Stressors and Cortisol Responses: A Theoretical Integration and Synthesis of Laboratory Research. *Psychological Bulletin*, *130*(3), 355–391. <https://doi.org/10.1037/0033-2909.130.3.355>
- Edwards, K. M., Bosch, J. A., Engeland, C. G., Cacioppo, J. T., & Marucha, P. T. (2010). Elevated Macrophage Migration Inhibitory Factor (MIF) is associated with depressive symptoms, blunted cortisol reactivity to acute stress, and lowered morning cortisol. *Brain, Behavior, and Immunity*, *24*(7), 1202–1208. <https://doi.org/10.1016/j.bbi.2010.03.011>
- Elbau, I. G., Brücklmeier, B., Uhr, M., Arloth, J., Czamara, D., Spoormaker, V. I., Czisch, M., Stephan, K. E., Binder, E. B., & Sämann, P. G. (2018). The brain's hemodynamic response function rapidly changes under acute psychosocial stress in association with genetic and endocrine stress response markers. *Proceedings of the National Academy of Sciences*, 201804340. <https://doi.org/10.1073/pnas.1804340115>
- Epel, E. S., McEwen, B., Seeman, T., Matthews, K., Castellazzo, G., Brownell, K. D., Bell, J., & Ickovics, J. R. (2000). Stress and Body Shape: Stress-Induced Cortisol Secretion Is Consistently Greater Among Women With Central Fat. *Psychosomatic Medicine*, *62*(5), 623–632.
- Finn, E. S., Shen, X., Scheinost, D., Rosenberg, M. D., Huang, J., Chun, M. M., Papademetris, X., & Constable, R. T. (2015). Functional connectome fingerprinting: Identifying individuals based on patterns of brain connectivity. *Nature neuroscience*, *18*(11), 1664–1671. <https://doi.org/10.1038/nn.4135>
- Foss, B., & Dyrstad, S. M. (2011). Stress in obesity: Cause or consequence? *Medical Hypotheses*, *77*(1), 7–10. <https://doi.org/10.1016/j.mehy.2011.03.011>
- Garawi, F., Devries, K., Thorogood, N., & Uauy, R. (2014). Global differences between women and men in the prevalence of obesity: Is there an association with gender inequality? *European Journal of Clinical Nutrition*, *68*(10), Article 10. <https://doi.org/10.1038/ejcn.2014.86>
- Goldfarb, E. V., Seo, D., & Sinha, R. (2019). Sex differences in neural stress responses and correlation with subjective stress and stress regulation. *Neurobiology of Stress*, *11*, 100177. <https://doi.org/10.1016/j.ynstr.2019.100177>
- Gossett, E. W., Wheelock, M. D., Goodman, A. M., Orem, T. R., Harnett, N. G., Wood, K. H., Mrug, S., Granger, D. A., & Knight, D. C. (2018). Anticipatory stress associated with functional magnetic resonance imaging: Implications for psychosocial stress research. *International Journal of Psychophysiology*, *125*, 35–41. <https://doi.org/10.1016/j.ijpsycho.2018.02.005>
- Grady, C. L., Rieck, J. R., Nichol, D., Rodrigue, K. M., & Kennedy, K. M. (2021). Influence of sample size and analytic approach on stability and interpretation of brain-behavior correlations in task-related fMRI data. *Human Brain Mapping*, *42*(1), 204–219. <https://doi.org/10.1002/hbm.25217>
- Grunberg, N. E., & Straub, R. O. (1992). The role of gender and taste class in the effects of stress on eating. *Health Psychology: Official Journal of the Division of Health Psychology, American Psychological Association*, *11*(2), 97–100. <https://doi.org/10.1037//0278-6133.11.2.97>

- Harrell, C. S., Gillespie, C. F., & Neigh, G. N. (2016). Energetic stress: The reciprocal relationship between energy availability and the stress response. *Physiology & Behavior, 166*, 43–55. <https://doi.org/10.1016/j.physbeh.2015.10.009>
- Hedley, A. A., Ogden, C. L., Johnson, C. L., Carroll, M. D., Curtin, L. R., & Flegal, K. M. (2004). Prevalence of overweight and obesity among US children, adolescents, and adults, 1999–2002. *JAMA, 291*(23), 2847–2850. <https://doi.org/10.1001/jama.291.23.2847>
- Henze, G.-I., Konzok, J., Kreuzpointner, L., Bärthel, C., Giglberger, M., Peter, H., Streit, F., Kudielka, B. M., Kirsch, P., & Wüst, S. (2021). Sex-specific interaction between cortisol and striato-limbic responses to psychosocial stress. *Social Cognitive and Affective Neuroscience, 16*(9), 972–984. <https://doi.org/10.1093/scan/nsab062>
- Herhaus, B., Ullmann, E., Chrousos, G., & Petrowski, K. (2020). High/low cortisol reactivity and food intake in people with obesity and healthy weight. *Translational Psychiatry, 10*(1), Article 1. <https://doi.org/10.1038/s41398-020-0729-6>
- Herman, J. P., Ostrander, M. M., Mueller, N. K., & Figueiredo, H. (2005). Limbic system mechanisms of stress regulation: Hypothalamo-pituitary-adrenocortical axis. *Progress in Neuro-Psychopharmacology and Biological Psychiatry, 29*(8), 1201–1213. <https://doi.org/10.1016/j.pnpbp.2005.08.006>
- Het, S., & Wolf, O. T. (2007). Mood changes in response to psychosocial stress in healthy young women: Effects of pretreatment with cortisol. *Behavioral Neuroscience, 121*(1), 11. <https://doi.org/10.1037/0735-7044.121.1.11>
- Hodes, G. E., & Kropp, D. R. (2023). Sex as a biological variable in stress and mood disorder research. *Nature Mental Health, 1*(7), Article 7. <https://doi.org/10.1038/s44220-023-00083-3>
- Incollingo Rodriguez, A. C., Epel, E. S., White, M. L., Standen, E. C., Seckl, J. R., & Tomiyama, A. J. (2015). Hypothalamic-pituitary-adrenal axis dysregulation and cortisol activity in obesity: A systematic review. *Psychoneuroendocrinology, 62*, 301–318. <https://doi.org/10.1016/j.psyneuen.2015.08.014>
- Kajantie, E., & Phillips, D. I. W. (2006). The effects of sex and hormonal status on the physiological response to acute psychosocial stress. *Psychoneuroendocrinology, 31*(2), 151–178. <https://doi.org/10.1016/j.psyneuen.2005.07.002>
- Kang, H. (2013). The prevention and handling of the missing data. *Korean Journal of Anesthesiology, 64*(5), 402–406. <https://doi.org/10.4097/kjae.2013.64.5.402>
- Kirschbaum, C., Kudielka, B. M., Gaab, J., Schommer, N. C., & Hellhammer, D. H. (1999). Impact of gender, menstrual cycle phase, and oral contraceptives on the activity of the hypothalamus-pituitary-adrenal axis. *Psychosomatic Medicine, 61*(2), 154–162.
- Klein, S. L., & Flanagan, K. L. (2016). Sex differences in immune responses. *Nature Reviews Immunology, 16*(10), Article 10. <https://doi.org/10.1038/nri.2016.90>
- Knigge, K. M., & Hays, M. (1963). Evidence of Inhibitive Role of Hippocampus in Neural Regulation of ACTH Release. *Proceedings of the Society for Experimental Biology and Medicine, 114*(1), 67–69. <https://doi.org/10.3181/00379727-114-28587>
- Kogler, L., Gur, R. C., & Derntl, B. (2015). Sex differences in cognitive regulation of psychosocial achievement stress: Brain and behavior. *Human Brain Mapping, 36*(3), 1028–1042. <https://doi.org/10.1002/hbm.22683>
- Kogler, L., Seidel, E.-M., Metzler, H., Thaler, H., Boubela, R. N., Pruessner, J. C., Kryspin-Exner, I., Gur, R. C., Windischberger, C., Moser, E., Habel, U., & Derntl, B. (2017). Impact of self-esteem and sex on stress reactions. *Scientific Reports, 7*(1), Article 1. <https://doi.org/10.1038/s41598-017-17485-w>
- Kuhn, L., Noack, H., Wagels, L., Prothmann, A., Schulik, A., Aydin, E., Nieratschker, V., Derntl, B., & Habel, U. (2023). Sex-dependent multimodal response profiles to psychosocial stress. *Cerebral Cortex, 33*(3), 583–596. <https://doi.org/10.1093/cercor/bhac086>
- Kühnel, A., Czisch, M., Sämann, P. G., Brückl, T., Spooemaker, V. I., Erhardt, A., Grandi, N. C., Ziebulka, J., Elbau, I. G., Namendorf, T., Lucae, S., Binder, E. B., & Kroemer, N. B. (2022). Spatiotemporal Dynamics of Stress-Induced Network Reconfigurations Reflect Negative Affectivity. *Biological Psychiatry, 92*(2), 158–169. <https://doi.org/10.1016/j.biopsych.2022.01.008>

- Kühnel, A., Kroemer, N. B., Elbau, I. G., Czisch, M., Sämann, P. G., Walter, M., & Binder, E. B. (2020). Psychosocial stress reactivity habituates following acute physiological stress. *Human Brain Mapping, 41*(14), 4010–4023. <https://doi.org/10.1002/hbm.25106>
- Kunz-Ebrecht, S. R., Mohamed-Ali, V., Feldman, P. J., Kirschbaum, C., & Steptoe, A. (2003). Cortisol responses to mild psychological stress are inversely associated with proinflammatory cytokines. *Brain, Behavior, and Immunity, 17*(5), 373–383. [https://doi.org/10.1016/S0889-1591\(03\)00029-1](https://doi.org/10.1016/S0889-1591(03)00029-1)
- Lee, M. R., Cacic, K., Demers, C. H., Haroon, M., Heishman, S., Hommer, D. W., Epstein, D. H., Ross, T. J., Stein, E. A., Heilig, M., & Salmeron, B. J. (2014). Gender differences in neural–behavioral response to self-observation during a novel fMRI social stress task. *Neuropsychologia, 53*, 257–263. <https://doi.org/10.1016/j.neuropsychologia.2013.11.022>
- Macht, M., & Mueller, J. (2007). Immediate effects of chocolate on experimentally induced mood states. *Appetite, 49*(3), 667–674. <https://doi.org/10.1016/j.appet.2007.05.004>
- Marek, S., Tervo-Clemmens, B., Calabro, F. J., Montez, D. F., Kay, B. P., Hatoum, A. S., Donohue, M. R., Foran, W., Miller, R. L., Hendrickson, T. J., Malone, S. M., Kandala, S., Feczko, E., Miranda-Dominguez, O., Graham, A. M., Earl, E. A., Perrone, A. J., Cordova, M., Doyle, O., ... Dosenbach, N. U. F. (2022). Reproducible brain-wide association studies require thousands of individuals. *Nature, 603*(7902), Article 7902. <https://doi.org/10.1038/s41586-022-04492-9>
- Meule, A., Reichenberger, J., & Blechert, J. (2018). Development and preliminary validation of the Salzburg Stress Eating Scale. *Appetite, 120*, 442–448. <https://doi.org/10.1016/j.appet.2017.10.003>
- Newman, E., O'Connor, D. B., & Conner, M. (2007). Daily hassles and eating behaviour: The role of cortisol reactivity status. *Psychoneuroendocrinology, 32*(2), 125–132. <https://doi.org/10.1016/j.psyneuen.2006.11.006>
- Ng, M., Fleming, T., Robinson, M., Thomson, B., Graetz, N., Margono, C., Mullany, E. C., Biryukov, S., Abbafati, C., Abera, S. F., Abraham, J. P., Abu-Rmeileh, N. M. E., Achoki, T., AlBuhairan, F. S., Alemu, Z. A., Alfonso, R., Ali, M. K., Ali, R., Guzman, N. A., ... Gakidou, E. (2014). Global, regional, and national prevalence of overweight and obesity in children and adults during 1980–2013: A systematic analysis for the Global Burden of Disease Study 2013. *The Lancet, 384*(9945), 766–781. [https://doi.org/10.1016/S0140-6736\(14\)60460-8](https://doi.org/10.1016/S0140-6736(14)60460-8)
- Nieuwenhuizen, A. G., & Rutters, F. (2008). The hypothalamic-pituitary-adrenal-axis in the regulation of energy balance. *Physiology & Behavior, 94*(2), 169–177. <https://doi.org/10.1016/j.physbeh.2007.12.011>
- Onat, A., Karadeniz, Y., Tusun, E., Yüksel, H., & Kaya, A. (2016). Advances in understanding gender difference in cardiometabolic disease risk. *Expert Review of Cardiovascular Therapy, 14*(4), 513–523. <https://doi.org/10.1586/14779072.2016.1150782>
- Ordaz, S., & Luna, B. (2012). Sex differences in physiological reactivity to acute psychosocial stress in adolescence. *Psychoneuroendocrinology, 37*(8), 1135–1157. <https://doi.org/10.1016/j.psyneuen.2012.01.002>
- Pasquali, R., Vicennati, V., Gambineri, A., & Pagotto, U. (2008). Sex-dependent role of glucocorticoids and androgens in the pathophysiology of human obesity. *International Journal of Obesity, 32*(12), Article 12. <https://doi.org/10.1038/ijo.2008.129>
- Preiss, K., Brennan, L., & Clarke, D. (2013). A systematic review of variables associated with the relationship between obesity and depression. *Obesity Reviews, 14*(11), 906–918. <https://doi.org/10.1111/obr.12052>
- Pruessner, J. C., Dedovic, K., Khalili-Mahani, N., Engert, V., Pruessner, M., Buss, C., Renwick, R., Dagher, A., Meaney, M. J., & Lupien, S. (2008). Deactivation of the Limbic System During Acute Psychosocial Stress: Evidence from Positron Emission Tomography and Functional Magnetic Resonance Imaging Studies. *Biological Psychiatry, 63*(2), 234–240. <https://doi.org/10.1016/j.biopsych.2007.04.041>
- Rabasa, C., & Dickson, S. L. (2016). Impact of stress on metabolism and energy balance. *Current Opinion in Behavioral Sciences, 9*, 71–77. <https://doi.org/10.1016/j.cobeha.2016.01.011>

- Rechlin, R. K., Splinter, T. F. L., Hodges, T. E., Albert, A. Y., & Galea, L. A. M. (2022). An analysis of neuroscience and psychiatry papers published from 2009 and 2019 outlines opportunities for increasing discovery of sex differences. *Nature Communications*, *13*(1), Article 1. <https://doi.org/10.1038/s41467-022-29903-3>
- Rohleder, N. (2019). Stress and inflammation – The need to address the gap in the transition between acute and chronic stress effects. *Psychoneuroendocrinology*, *105*, 164–171. <https://doi.org/10.1016/j.psyneuen.2019.02.021>
- Scheinost, D., Noble, S., Horien, C., Greene, A. S., Lake, E. MR., Salehi, M., Gao, S., Shen, X., O'Connor, D., Barron, D. S., Yip, S. W., Rosenberg, M. D., & Constable, R. T. (2019). Ten simple rules for predictive modeling of individual differences in neuroimaging. *NeuroImage*. <https://doi.org/10.1016/j.neuroimage.2019.02.057>
- Schmalenberger, K. M., Tauseef, H. A., Barone, J. C., Owens, S. A., Lieberman, L., Jarczok, M. N., Girdler, S. S., Kiesner, J., Ditzen, B., & Eisenlohr-Moul, T. A. (2021). How to study the menstrual cycle: Practical tools and recommendations. *Psychoneuroendocrinology*, *123*, 104895. <https://doi.org/10.1016/j.psyneuen.2020.104895>
- Shen, X., Tokoglu, F., Papademetris, X., & Constable, R. T. (2013). Groupwise whole-brain parcellation from resting-state fMRI data for network node identification. *NeuroImage*, *0*, 403–415. <https://doi.org/10.1016/j.neuroimage.2013.05.081>
- Stephens, M. A. C., Mahon, P. B., McCaul, M. E., & Wand, G. S. (2016). Hypothalamic-pituitary-adrenal axis response to acute psychosocial stress: Effects of biological sex and circulating sex hormones. *Psychoneuroendocrinology*, *66*, 47–55. <https://doi.org/10.1016/j.psyneuen.2015.12.021>
- Sui, J., Jiang, R., Bustillo, J., & Calhoun, V. (2020). Neuroimaging-based Individualized Prediction of Cognition and Behavior for Mental Disorders and Health: Methods and Promises. *Biological Psychiatry*, *88*(11), 818–828. <https://doi.org/10.1016/j.biopsych.2020.02.016>
- Thorand, B., Baumert, J., Döring, A., Herder, C., Kolb, H., Rathmann, W., Giani, G., & Koenig, W. (2006). Sex differences in the relation of body composition to markers of inflammation. *Atherosclerosis*, *184*(1), 216–224. <https://doi.org/10.1016/j.atherosclerosis.2005.04.011>
- Tomiyaama, A. J., Dallman, M. F., & Epel, E. S. (2011). Comfort food is comforting to those most stressed: Evidence of the chronic stress response network in high stress women. *Psychoneuroendocrinology*, *36*(10), 1513–1519. <https://doi.org/10.1016/j.psyneuen.2011.04.005>
- Tramunt, B., Smati, S., Grandgeorge, N., Lenfant, F., Arnal, J.-F., Montagner, A., & Gourdy, P. (2020). Sex differences in metabolic regulation and diabetes susceptibility. *Diabetologia*, *63*(3), 453–461. <https://doi.org/10.1007/s00125-019-05040-3>
- Varghese, M., Griffin, C., & Singer, K. (2017). The Role of Sex and Sex Hormones in Regulating Obesity-Induced Inflammation. In F. Mauvais-Jarvis (Hrsg.), *Sex and Gender Factors Affecting Metabolic Homeostasis, Diabetes and Obesity* (S. 65–86). Springer International Publishing. https://doi.org/10.1007/978-3-319-70178-3_5
- Varoquaux, G., Raamana, P. R., Engemann, D. A., Hoyos-Ildrobo, A., Schwartz, Y., & Thirion, B. (2017). Assessing and tuning brain decoders: Cross-validation, caveats, and guidelines. *NeuroImage*, *145*, 166–179. <https://doi.org/10.1016/j.neuroimage.2016.10.038>
- Zellner, D. A., Loaiza, S., Gonzalez, Z., Pita, J., Morales, J., Pecora, D., & Wolf, A. (2006). Food selection changes under stress. *Physiology & Behavior*, *87*(4), 789–793. <https://doi.org/10.1016/j.physbeh.2006.01.014>
- Zschucke, E., Renneberg, B., Dimeo, F., Wüstenberg, T., & Ströhle, A. (2015). The stress-buffering effect of acute exercise: Evidence for HPA axis negative feedback. *Psychoneuroendocrinology*, *51*, 414–425. <https://doi.org/10.1016/j.psyneuen.2014.10.019>

REVIEWERS' COMMENTS:

Reviewer #1 (Remarks to the Author):

The authors have adequately addressed my concerns and the manuscript has been improved. No further major comments.

(Minor)

- Some characters are too small in Supplementary Figure 1. Please magnify them.

Reviewer #2 (Remarks to the Author):

I would like to thank the reviewers for incorporating the feedback, and have no further comments.

Reviewer #3 (Remarks to the Author):

The authors have addressed all of my concerns. I would recommend this article for publication.

We thank the reviewers for their feedback and their positive evaluation of the revised manuscript.

Reviewer #1 (Remarks to the Author):

The authors have adequately addressed my concerns and the manuscript has been improved. No further major comments.

(Minor)

1) Some characters are too small in Supplementary Figure 1. Please magnify them.

We thank the reviewer for pointing out that the text is too small within a figure. We have changed the figure so that all labels are bigger.

Reviewer #2 (Remarks to the Author):

I would like to thank the reviewers for incorporating the feedback, and have no further comments.

Reviewer #3 (Remarks to the Author):

The authors have addressed all of my concerns. I would recommend this article for publication.